# A General-Purpose Multi-Modal OOD Detection Framework

**Viet Duong**                                                          *vqduong@wm.edu*
*Department of Computer Science, William & Mary*

**Qiong Wu**                                                            *qw6547@att.com*
*AT&T Labs*

**Zhengyi Zhou**                                                        *zz547k@att.com*
*AT&T Labs*

**Eric Zavesky**                                                        *ez2685@att.com*
*AT&T Labs*

**Wen-Ling Hsu**                                                        *wenlhsu@gmail.com*
*AT&T Labs*

**Han Zhao**                                                            *hanzhao@illinois.edu*
*Department of Computer Science, University of Illinois at Urbana-Champaign*

**Huajie Shao**\*                                                       *hshao@wm.edu*
*Department of Computer Science, William & Mary*

**Reviewed on OpenReview:** *https://openreview.net/forum?id=nYzws7sSzo*

## Abstract

Out-of-distribution (OOD) detection seeks to identify test samples that deviate from the training data, which is critical to ensuring the safety and reliability of machine learning (ML) systems. While a plethora of methods have been developed to detect uni-modal OOD samples, only a few have focused on multi-modal OOD detection. Current contrastive learning-based methods primarily address multi-modal OOD detection in a scenario where an image is not related to the class labels in training data. However, ML systems in the real-world applications may encounter a broader spectrum of anomalies caused by different factors like systematic errors in labeling, environmental changes, and sensor malfunctions. Hence, we propose a new method to be able to simultaneously detect anomalies from multiple different OOD scenarios, arising from fine-grained image features and textual descriptions, instead of large categorical information. To achieve this goal, we propose a general-purpose weakly-supervised OOD detection framework, called WOOD, that combines a binary classifier and a contrastive learning module to reap the benefits of both. In order to better distinguish in-distribution (ID) samples from OOD ones, we employ the Hinge loss to constrain the similarity of their latent representations. Moreover, we devise a new scoring metric that fuses predictions from both the binary classifier and contrastive learning to enhance OOD detection. Extensive experimental results on multiple benchmarks demonstrate that the proposed WOOD significantly outperforms the state-of-the-art methods for multi-modal OOD detection. Importantly, our approach can achieve superior detection performance in a variety of OOD scenarios.

---

\*Corresponding author.

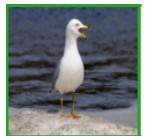 a small yellow colored bird with a black head and body strips and a small beak

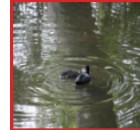 two black birds are floating around in the river one looks curious about the other mouth

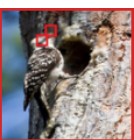 medium size bird with brown and white speckled wings white

(a) Misaligned - an ID image not aligned with its textual description (e.g., the textual description belongs to another class).

(b) New-domain - an aligned image-text pair from a dataset of another domain (e.g., containing two birds instead of one).

(c) Noisy - an ID and aligned image-caption pair but with blurry patches (highlighted in red squares) in the image (sensory faults).

Figure 1: Examples of three OOD scenarios in CUB-200 dataset, with the textual descriptions for each scenario. Images are indicated as OOD with red border, and ID with green. Regarding the text modality, corresponding textual descriptions are in red for OODs and green for IDs.

## 1 Introduction

Out-of-distribution (OOD) detection (Yang et al., 2021; Bogdoll et al., 2022; Ming et al., 2022c; Ruff et al., 2021; Ma et al., 2021) aims at identifying whether a test sample differs from the training data. Such detection is crucial for ensuring the safety and reliability of machine learning (ML) systems (Wang et al., 2020; Hussain & Zeadally, 2018), such as autonomous driving and AI diagnosis (Davenport & Kalakota, 2019; Han et al., 2020; MacDonald et al., 2022). Due to its importance, OOD detection has received a lot of attention from both industry and academia.

While extensive studies have focused on single-modal OOD detection (Zadorozhny et al., 2022; Roy et al., 2022; Narayanaswamy et al., 2023), multi-modal OOD detection remains underexplored. In reality, multi-modal OOD detection is particularly important to many high-stakes applications like clinical diagnosis and autonomous driving. These applications often fuse information from different data sources and modalities to improve the reliability of decision making. For instance, a majority of AI diagnosis systems integrate multi-modal data, such as radiology imaging (X-ray, ultrasound, CT, MRI, etc.), physician's annotations, and electronic health records, to enhance their prediction performance. In the real-world applications, those multi-modal ML systems inevitably encounter "outlier" conditions caused by various factors, such as machine defects, sensor faults, and operator errors. Hence, a few recent works (Ming et al., 2022b; Esmaeilpour et al., 2022) have explored multi-modal visual-language OOD detection, but they narrow their scope to a special scenario where images come from a new domain. Moreover, these work often employ CLIP-based techniques (Ming et al., 2022a; Miyai et al., 2023; Tao et al., 2023; Miyai et al., 2024) to identify whether a query image matches one of the training set's *class-level labels* (e.g., pneumonia). As a result, these methods fail to identify OOD scenarios such as misaligned pairings of an image and its *fine-grained textual description*.

The goal of this work is to develop a general-purpose multi-modal OOD detection model that can identify OOD in various scenarios in a fine-grained manner. We focus on three different OOD scenarios for multi-sensory data, as illustrated in Fig 1: (1) *Misaligned* - unaligned pairs of data samples, e.g., an in-distribution (ID) image is not aligned with its textual information; (2) *New-domain* - aligned pairs of data samples collected from a new domain, e.g., aligned images and text from a new environment with a different distribution from training; (3) *Noisy* - partially noisy data samples, e.g., images coming from the same environment but with blurry patches. The research question is: How can we detect OOD samples from all these OOD scenarios simultaneously? Existing OOD detection approaches, such as the CLIP-based methods (Ming et al., 2022a; Esmaeilpour et al., 2022; Fort et al., 2021) and weakly-supervised classifications (Ji et al., 2020; Wang et al., 2021), only focus on one of these scenarios (*new-domain* or *noisy*), thus failing to generalize to all of them. Fig. 2 illustrates a motivating example of multi-modal OOD detection on CUB-200 dataset (Wah et al., 2011) using our weakly-supervised CLIP-based contrastive learning method and the binary classifier. We can observe from Fig. 2 (a) that weakly-supervised CLIP can only detect *misaligned* OODs, but is not effective in *new-domain* or *noisy* scenarios. Specifically, weakly-supervised CLIP is not effective for *new-domain* OODs because we deliberately create a more difficult *new-domain* scenario by choosing OODs as samples labeled with "bird" from COCO data (Lin et al., 2014), which has *the same label but different textual information* as those ID samples in the training set of CUB-200, instead of samples with other classes (e.g., person, dog, etc.) in previous works (Ming et al., 2022a; Zhou et al., 2022a; Miyai et al., 2023). In

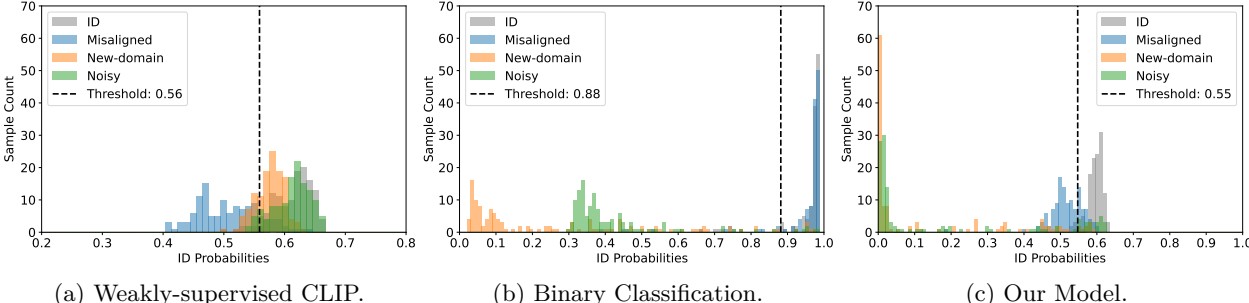

Figure 2: A motivating example of multi-modal OOD detection on CUB-200 dataset. Following prior work (Ming et al., 2022b), we choose a threshold that includes 95% of in-distribution test samples (gray), and identify a test sample as OOD if its prediction score is *below that threshold*, i.e., to the left of the dashed line. If an OOD detector works effectively, the prediction scores for all samples in all three OOD scenarios should be concentrated to the left of the dashed line. We can observe from (a) that weakly-supervised contrastive learning can only detect *misaligned* OOD samples (blue), but fails to detect *new-domain* and *noisy* OODs. Conversely, a binary classifier in (b) can identify OOD samples in *new-domain* (orange) and *noisy* (green) scenarios, but it does not work for *misaligned* test samples. Motivated by these observations, we develop a general-purpose OOD detection model (c) that can effectively detect OOD samples from all three scenarios.

contrast, a binary classifier can only identify *new-domain* or *noisy* OOD samples, but fails in the *misaligned* scenario, as illustrated in Fig. 2 (b). This observation motivates us to develop a new OOD detection model that combines both approaches to complement each other.

**Our Contributions.** We propose the WOOD, a weakly-supervised multi-modal OOD detection model that can identify three OOD scenarios at the same time. As illustrated in Fig. 3, the proposed WOOD is composed of two components: a binary classifier for classifying OOD samples and a contrastive learning module for measuring the similarity scores between multiple data modalities. On the contrastive learning side, we adopt the Hinge loss to maximize the similarity scores of ID samples and minimize those of OOD samples to better distinguish them. On the binary classifier side, we develop a novel Feature Sparsity Regularizer to better integrate important features from data of multiple modalities. Then a new scoring metric is designed to fuse the prediction results from these two components. Finally, we evaluate the proposed WOOD model on three real-world benchmark datasets. Experimental results demonstrate that our method can successfully identify OOD samples with a high accuracy under three different scenarios, which significantly outperforms the CLIP-based baselines.

## 2 Related Work

In this section, we review and contrast related work on OOD detection in machine learning. We group them into two categories: single-modal OOD detection and multi-modal OOD detection. Additionally, we further discuss related work on multi-modal OOD generalization in Appendix A.

**Single-Modal OOD Detection.** There exists a plethora of works on single-modal OOD detection (Salehi et al., 2022) for machine learning. They can be generally categorized into three types: (i) vision OOD detection, (ii) text OOD detection, and (iii) time-series OOD detection (Du et al., 2022). For vision OOD detection, various methods have been developed, including softmax confidence score (DeVries & Taylor, 2018; Hein et al., 2019; Hendrycks & Gimpel, 2017; Huang & Li, 2021), energy-based score (Yang et al., 2021; Sun et al., 2021; Sun & Li, 2022), distance-based method (Sun et al., 2022; Ren et al., 2021; Podolskiy et al., 2021; Techapanurak et al., 2020; Zaeemzadeh et al., 2021; Van Amersfoort et al., 2020), and generative models (Li et al., 2022b; Xiao et al., 2020). For instance, Liu et al. (2020) proposed an energy-based OOD detection method with theoretical analysis. Sun et al. (2022) developed nearest neighbors to improve the flexibility and generality of OOD detection. Yoo et al. (2023) considered data augmentation as a hyperparameter for generating pseudo-anomalies, and investigated its effectiveness for image-based self-supervised anomaly detection. For text OOD detection, pre-trained language models (Podolskiy et al., 2021; Zhou et al., 2021)

are commonly used due to their robustness in identifying OOD samples in natural languages. Other methods, such as data augmentation (Zhan et al., 2021) and contrastive learning (Zhou et al., 2022b; 2021), have also been developed for OOD detection. Furthermore, some researchers have focused on OOD detection in time-series data (Romero & Estévez, 2022; Kaur et al., 2022; Georgescu et al., 2021), where several ML models have been developed for video anomaly detection (Georgescu et al., 2021; Wang et al., 2018). Wang et al. (2018) combined LSTM with CNN to improve anomaly detection using a spatio-temporal auto-encoder. Li et al. (2022a) leveraged generative models to predict middle frames based on past and future frames. However, these methods mainly detect OOD samples using unimodal data, such as images or text. In contrast, we develop a general-purpose model that combines multi-modal data, such as images and textual information, to enhance the performance of OOD detection.

**Multi-Modal OOD Detection.** Some studies (Sun et al., 2020; Wang et al., 2021) have adopted multi-modal data to improve the OOD detection accuracy based on deep neural networks (DNNs). (Wang et al., 2021) proposed a multi-modal transformer network that combines Radar and LiDAR data to detect radar ghost targets. Ji et al. (2020) developed a supervised VAE (SVAE) model that integrates sensor data of multiple modalities to detect an anomalous operation mode of the car. To improve the accuracy of detecting abnormal driving segments, Qiu et al. (2022) developed an unsupervised contrastive approach that uses generative adversarial networks to extract latent features from five modalities. More recently, CLIP-based methods (Ming et al., 2022a; Esmaeilpour et al., 2022; Fort et al., 2021; Jeong et al., 2023) have been developed to detect OOD samples. (Esmaeilpour et al., 2022) designed a zero-shot OOD detection model based on pre-trained CLIP (Radford et al., 2021) to detect and generate candidate labels for test images of unknown classes. However, this method heavily relies on a set of candidate labels. To overcome this issue, Ming et al. (2022a) developed a zero-shot method called Maximum Concept Matching (MCM) based on a pre-train CLIP model, that can detect visual OOD samples, i.e., OODs arising from the image modality. Miyai et al. (2023) further introduced Global-Local(GL)-MCM, which utilizes local CLIP visual features to enable zero-shot OOD detection for background-related OOD features. Additionally, Jeong et al. (2023) developed WinCLIP to detect multiple scenarios of visual anomaly by matching multi-scale CLIP image embeddings obtained from image windows of varying sizes with text prompts describing the anomaly (e.g., "a blurry photo"), but known anomaly prompts are required to achieve high zero-shot detection accuracy. Furthermore, latest research explored various fine-tuning strategies for CLIP, such as prompt learning (Zhou et al., 2022a; Miyai et al., 2024) and outlier synthesis (Tao et al., 2023), for enhanced OOD detection based on MCM and GL-MCM. While these methods have shown good performance on multi-modal OOD detection, they can only detect visual OOD samples rather than both an image and its associated textual description. Hence, they are not applicable to other scenarios we are exploring, such as misalignment and noisy samples.

Different from prior works, we develop a general-purpose multi-modal OOD detector that can identify OOD samples arising from three different scenarios in a fine-grained manner. Our proposed method leverages both weakly supervised learning and contrastive learning for improving OOD detection.

## 3  Preliminaries

This section begins by introducing the problem of detecting OOD samples in multi-modal data. Subsequently, we delve into the applications of contrastive learning for multi-modal OOD detection

### 3.1  Problem Statement

We consider the problem of detecting multi-modal OOD samples under three different scenarios, as mentioned in Sec. 1. In this paper, we use vision-language modeling as a running problem for multi-modal OOD detection. Given a batch of $N$ pairs of images and texts, along with their positive ID labels, denoted by $\{(x_n, t_n), y_n = 1\}_{n=1}^N$, we generate a small number of OOD samples $K$ prior to training. Specifically, we select $K \ll N$ pairs of images and texts using a random seed, and convert them to OODs according to the three scenarios above, such that $\{(x_k, t_k), y_k = 0 | k \in \{k_1, \ldots, k_K\} \subset \{1, \ldots, N\}\}$, and the remaining $N - K$ ID pairs are unchanged. The goal of this work is to improve the OOD detection performance for multiple scenarios using weak supervision on a small number of generated OOD samples.

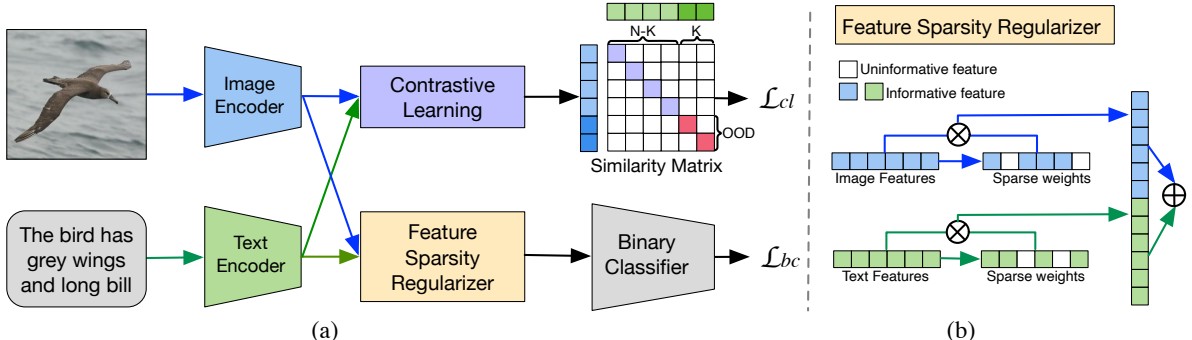

Figure 3: (a) The overall framework of the WOOD detector. It consists of two main parts: 1) a constrastive learning module and 2) a binary classifier. Contrastive learning aims to maximize the difference in similarity scores between ID and OOD samples while the binary classifier is used to predict the probability of being OOD. (b) An expanded detailed view of the Feature Sparsity Regularizer in the WOOD framework. This regularizer integrates important features from data of multiple modalities to improve classification accuracy.

## 3.2 Contrastive Learning

Contrastive learning aims to encode pairs of data samples into latent representations by making similar samples close to each other and dissimilar ones far apart. One well-known method is the vision-language pre-trained model CLIP, which jointly trains an image encoder and a text encoder to learn the latent representations from text paired with images using zero-shot learning. Specifically, it encodes a pair of image $x_n$ and text $t_n$ into the latent representations $\mathcal{I}(x_n)$ and $\mathcal{T}(t_n)$, respectively, and then adopts cosine similarity to minimize the distance of their representations $\mathcal{I}(x_n)$ and $\mathcal{T}(t_n)$. Due to its excellent performance in learning the latent representations of images and texts, a few CLIP-based methods have been developed to detect multi-modal OOD samples. However, existing methods can only detect one specific scenario where a given image and its textual information come from a new domain with a distribution shift. In the following section, we develop a general-purpose OOD detection framework to identify OODs arising from several different scenarios.

## 4 Proposed Model

In this section, we describe the proposed multi-modal OOD detection framework and then devise a new scoring metric for detecting OODs under three different scenarios.

We propose a weakly-supervised OOD detector, called WOOD, that combines a classifier and contrastive learning to simultaneously detect three different OOD scenarios mentioned in Section 1. Fig. 3 illustrates the overall multi-modal OOD detection framework that consists of two components: a contrastive learning module and a binary classifier. The core idea is to use contrastive learning to learn the representations of different data modalities by enforcing the similarity scores of ID pairs to be higher than those of OOD samples. Then WOOD combines the similarity scores from contrastive learning and the prediction results from a binary classifier to identify OOD samples. Below, we detail the two main components of the proposed WOOD method.

**Contrastive Learning with Hinge Loss**. Inspired by CLIP-based OOD detection methods, we adopt an image encoder and a text encoder to learn the representations of input pairs (image and text) using contrastive learning. However, unlike existing zero-shot CLIP-based detectors, we add a small number of OOD samples to better separate the representations of ID and OOD samples. Thus, our goal is to maximize the cosine similarity of representations learned from ID samples but and minimize those of OOD samples. Let $(x_n, t_n)$ be a pair of input image and text, and their corresponding representations denote $(\mathcal{I}(x_n), \mathcal{T}(t_n))$. Then, the CLIP contrastive loss (Radford et al., 2021) can be modified to maximize the *cross entropy* of the

cosine similarity between labeled OOD pairs and minimize that of ID pairs, which is given by:

$$\mathcal{L}_1 = -\frac{1}{N-K} \sum_{n=1}^{N-K} \log \frac{e^{S_{id}(x_n^+, t_n^+)}}{\sum_{i=1}^{N-K} e^{S_{id}(x_n^+, t_i^+)}} + \frac{1}{K} \sum_{k=1}^{K} \log \frac{e^{S_{ood}(x_k^-, t_k^-)}}{\sum_{i=1}^{K} e^{S_{ood}(x_k^-, t_i^-)}}, \tag{1}$$

where $S_{id}(x_n^+, t_n^+) = \frac{\mathcal{I}(x_n^+) \cdot \mathcal{T}(t_n^+)}{\|\mathcal{I}(x_n^+)\| \|\mathcal{I}(t_n^+)\|}$ and $S_{ood}(x_k^-, t_k^-) = \frac{\mathcal{I}(x_k^-) \cdot \mathcal{T}(t_k^-)}{\|\mathcal{I}(x_k^-)\| \|\mathcal{I}(t_k^-)\|}$ represent the cosine similarity between image and text features for ID and OOD pairs, respectively. Here "+" and "-" mean the sample is from ID and OOD, respectively. In addition, $N - K$ and $K$ respectively denote the number of ID and OOD pairs.

To further maximize the difference in latent representations between ID and OOD samples, we adopt Hinge loss instead of $\mathcal{L}_1$ in Eq. 1 to constrain their cosine similarity. In this work, we consider Hinge loss for both ID pairs and labeled OOD pairs in the objective function.

First, we introduce Hinge loss for $N - K$ ID samples, as shown in the upper part of the similarity matrix in Fig 3. It is given by:

$$\mathcal{L}_{id} = \sum_{n=1}^{N-K} \left( \frac{1}{N} \sum_{i=1, i \neq n}^{N} \max \left(0, m - S_{id}(x_n^+, t_n^+) + S_{id}(x_n^+, t_i^{+/-})\right) \right), \tag{2}$$

where $m$ is a margin, $S_{id}(x_n^+, t_n^+)$ represents the cosine similarity of $N-K$ aligned ID pairs and $S_{id}(x_n^+, t_i^{+/-})$ represents the cosine similarity between $N - K$ ID images and all $N$ texts (including OOD samples), where each text either does not align with its corresponding ID image or belongs to an OOD sample. In short, the objective, $\mathcal{L}_{id}$, aims to maximize the difference between the aligned ID pairs and incorrect pairings by ensuring that such difference is larger than a margin $m$.

Second, we introduce Hinge loss for $K$ OOD samples to constrain their cosine similarity to a small value. Hence, we have:

$$\mathcal{L}_{ood} = \sum_{k=1}^{K} \left( \frac{1}{N} \sum_{i=1}^{N} \max \left(0, -m + S_{ood}(x_k^-, t_i^{+/-})\right) \right), \tag{3}$$

where $S_{ood}(x_k^-, t_i^{+/-})$ represents the cosine similarity between each of $K$ OOD images and all $N$ texts (including OOD samples). By definition, the $K$ OOD images should not align with any of the $N - K$ ID texts, since each OOD image is not aligned with its corresponding text as described in the *misaligned* scenario. Additionally, we should discourage the model from aligning *new-domain* and *noisy* OOD pairs.

By combining the above loss functions, $\mathcal{L}_{id}$ and $\mathcal{L}_{ood}$, for ID and OOD samples, the overall contrastive loss is given by:

$$\mathcal{L}_{cl} = \frac{1}{N} \left( \mathcal{L}_{id} + \mathcal{L}_{ood} \right). \tag{4}$$

During inference, inspired by Wang et al. (2023a), we apply the sigmoid function $\sigma$ to the cosine similarity to obtain the ID probabilities:

$$P_{cl} = \sigma \left( \frac{S(x_n, t_n)}{\tau} \right), \tag{5}$$

where $S(x_n, t_n)$ denotes the cosine similarity, and $\tau \leq 1$ is the temperature for scaling the cosine similarity. Note that as the cosine similarity is in the range $[-1, 1]$, when we set a small value of $\tau$ as 0.1 in our experiments, the range of $P_{cl}$ is close to $(0, 1)$.

**Binary Classifier**. We further adopt weakly-supervised learning to classify OOD samples since recent studies have illustrated that it can significantly outperform unsupervised learning methods by adding some OOD samples (Tian et al., 2020; Sultani et al., 2018; Majhi et al., 2021). However, the challenge lies in how to integrate image and text features for improved classification accuracy. One naive method is to concatenate their embeddings directly and then feed them into a classifier. But this simple fusion approach does not perform well since the informativeness of different features may vary for different samples (Jiang

---

**Algorithm 1** The Proposed WOOD Model

---

1: **Input:** A batch of $N$ pairs of images and texts, with $N - K$ ID pairs and $K$ labeled OOD pairs.
2: **Output:** Samples labeled as OOD or ID.
3: Encode ID pairs into $\mathcal{I}(x_n^+)$ and $\mathcal{T}(t_n^+)$.
4: Encode OOD pairs into $\mathcal{I}(x_n^-)$ and $\mathcal{T}(t_n^-)$.
5: Compute Hinge loss on cosine similarity for ID samples according to Eq. 2 and OOD samples according to Eq. 3 for contrastive learning.
6: Compute the total contrastive loss $\mathcal{L}_{cl}$ in Eq. 4.
7: Compute loss of the binary classifier $\mathcal{L}_{bc}$ in Eq. 6.
8: Jointly train the binary classifier and contrastive learning based on the overall objective in Eq. 7.
9: Identify OOD samples based on the designed scoring metric in Eq. 8.

---

et al., 2023; Han et al., 2022). To address this issue, we develop a feature sparsity regularizer to select and integrate important features from data of the two modalities, as illustrated in Fig. 3(b). Specifically, we train single-layer encoder MLP networks $E^{\mathcal{I}} : \mathcal{I}(x_n) \to w_n^{\mathcal{I}}$ and $E^{\mathcal{T}} : \mathcal{T}(t_n) \to w_n^{\mathcal{T}}$ to use features obtained from image encoder and text encoders. Here, the dimension of $w_n^{\mathcal{I}}$ matches that of $x_n$, and the dimension of $w_n^{\mathcal{T}}$ matches that of $t_n$. These weight features are updated by a sigmoid activation $\sigma$ to assign higher weights to informative features and lower weights to the uninformative features. Namely, the weight vectors $\sigma(w_n^{\mathcal{I}})$ and $\sigma(w_n^{\mathcal{T}})$ are multiplied element-wise (denoted by $\otimes$) with $\mathcal{I}(x_n)$ and $\mathcal{T}(t_n)$ respectively. After that, we fuse the features from two modalities by concatenation ($\oplus$), yielding $h_n = \oplus[\mathcal{I}(x_n) \otimes \sigma(w_n^{\mathcal{I}}), \mathcal{T}(t_n) \otimes \sigma(w_n^{\mathcal{T}})]$. Next, we adopt a binary classifier to identify ID/OOD samples based on the fused features $h_n$. To introduce sparsity in the weight vectors, we use $L_1$ normalization $\|\sigma(w_n^{\mathcal{I}})\|_1$ and $\|\sigma(w_n^{\mathcal{T}})\|_1$ and add them to our binary cross entropy (BCE) loss function, which is given by:

$$\mathcal{L}_{bc} = \frac{1}{N} \left( \sum_{i=1}^{N} \text{BCE}(y_i, \hat{y}_i) + \|\sigma(w_n^{\mathcal{I}})\|_1 + \|\sigma(w_n^{\mathcal{T}})\|_1 \right) \tag{6}$$

where $\text{BCE}(y_i, \hat{y}_i) = -(y_i \log(\hat{y}_i) + (1 - y_i) \log(1 - \hat{y}_i))$. Note that here $y_i = 0$ means the $i$-th test sample is OOD while it is an ID sample when $y_i = 1$.

**Overall objective**. Finally, we jointly train the contrastive learning module and binary classifier for OOD detection. The overall objective for multi-OOD detection is given by:

$$\mathcal{L} = (1 - \lambda)\mathcal{L}_{cl} + \lambda\mathcal{L}_{bc}, \tag{7}$$

where $\mathcal{L}_{bc}$ is the contrastive loss, $\mathcal{L}_{bc}$ is the binary cross-entropy loss for the classifier, and $\lambda \in [0, 1]$ is the weight for balancing the two respective terms.

### 4.1 New Scoring Metric

To improve the performance of OOD detection during inference, we introduce a novel scoring metric that combines predictions from the contrastive learning module and the binary classifier. The key insight is that we identify an image and text pair as ID only when both the contrastive learning and binary classifier predict that sample as ID. In all other cases, we identify it as OOD. We codify this condition with the following scoring metric for identifying OOD samples:

$$P_{ood} = 1 - P_{bc}P_{cl}, \tag{8}$$

where $P_{bc}$ and $P_{cl}$ denote the ID probabilities in $(0, 1)$ from the binary classifier and contrastive learning, respectively. Subsequent sections demonstrate that the new scoring metric can help detect OOD samples in all three OOD scenarios effectively.

## 4.2 Summary of Proposed Model

We summarize the proposed WOOD model in Algorithm 1. The basic idea is to map ID and OOD pairs (image and text) into latent representations, and then calculate their cosine similarity. Then, we use Hinge loss to maximize the difference in similarity scores between ID and OOD samples. Moreover, we feed the fused latent representations into a binary classifier to classify ID or OOD samples. Finally, we jointly train the binary classifier and the contrastive learning component for OOD detection.

## 5 Experiments

In this section, we carry out extensive experimentation to evaluate the performance of the proposed WOOD model on multiple benchmark datasets. Then, we conduct ablation studies to explore how the main components in model design and hyperparameters impact OOD detection performance.

### 5.1 Datasets

We implement experiments on the three real-world datasets: CUB-200 (Wah et al., 2011), MIMIC-CXR (Johnson et al., 2019), and COCO (Lin et al., 2014). For CUB-200, the textual information comes from literature (Reed et al., 2016). COCO and MIMIC-CXR contain images and their textual descriptions.

**Three OOD scenarios**. We generate three different OOD scenarios using the above datasets as follows.

- **Misaligned**. Randomly select a subset of ID images and their textual description from a given dataset and shuffle them so that each image is not aligned with its corresponding textual information. Specifically, we select images from one category and the unaligned textual descriptions from another category, ensuring that each pair of OOD images and text are not aligned.
- **New-domain**. Choose OOD samples from another new dataset different from the training data. For instance, when conducting experiments on MIMIC-CXR data, we select some pairs of texts and images from Indiana University (IU) X-ray (Demner-Fushman et al., 2016) as OOD samples.
- **Noisy**. Add Gaussian blur (whitening for X-ray images) to several patches in the ID images, such that each image is partially blurry but its corresponding textual information is correct. Additionally, we evaluate CLIP's OOD generalization to noisy images on the CUB-200 dataset in Appendix B.

Table 1 summarizes the detailed information about generating three OOD scenarios using the above datasets in the experiments.

Table 1: Detailed Summary of Our Three OOD Scenarios.

| Scenarios | CUB-200 | MIMIC-CXR | COCO |
|---|---|---|---|
| Misaligned | Randomly select pairs of ID images and texts, swap their textual descriptions with textual descriptions of images from different categories, and label these misaligned pairs as OOD. | | |
| New-domain | Samples from COCO with birds | IU X-ray (Demner-Fushman et al., 2016) | GCC (Sharma et al., 2018) |
| Noisy | Randomly select pairs of ID images and texts, add a small number of blurry/whitened patches to each, and label these blurry pairs as OOD. | | |

### 5.2 Model Configurations

Following prior CLIP-based detectors, we also use the CLIP model (ViT-B/16 (Radford et al., 2021)) as the backbone of the contrastive learning module. The two encoders are $CLIP_{image}$ and $CLIP_{text}$, which are pre-trained Transformer models for image and text (Radford et al., 2021) respectively. We do not change the base encoders but fine-tune them with Hinge loss in Eq. (4) for both feature alignment and OOD detection. Recall that WOOD also has a Feature Sparsity Regularizer module (Figure 3), which is a single projection layer (MLP) with a sigmoid activation. We set its hidden size to 512, the same as the dimensions of the output embeddings from $CLIP_{image}$ and $CLIP_{text}$. The Binary Classifier is a 3-layer fully connected network with ReLU activation, which outputs a single probability score for binary OOD classification and the layer hidden size is 1024, 512, and 128 respectively. We train the proposed WOOD model using Adam optimizer (Kingma & Ba, 2014) with initial learning rate $5e^{-6}$ and stepped learning rate schedule. Additionally, the batch size is set to 128 in all experiments, and $\lambda = 0.8$ for the overall training adjective in Eq. (7). Regarding

the margin of the Hinge loss, we choose $m = 0.2$ for CUB-200 and MIMIC-CXR, and $m = 0.3$ for COCO after grid search. Note that subsequent ablation studies in Section 5.5 explore the impact of these hyper-parameters on detection performance. To improve detection performance on three scenarios simultaneously, we randomly select 1% of the training data and replace them with labeled OOD samples for each scenario, as described in Tab. 1, resulting in a total of 3% OOD samples and 97% of ID samples for weakly-supervised training. During evaluation, we use the same ratio (25%) of test samples for ID and three OOD scenarios. Finally, regarding inference, following previous research (Ming et al., 2022b), we choose a threshold $\delta$ (e.g., 0.6) at inference so that a high fraction of ID data is above the threshold (i.e., OOD samples are identified when $P_{ood} > 1 - \delta$). Following prior work (Ming et al., 2022b), we report the false positive rate for 95% of recall of ID samples, FPR95, to measure the prediction error when OODs are identified by threshold $\delta$. Additionally, we report the AUROC scores, which is a threshold-free OOD detection metric commonly used in the literature (Hendrycks & Gimpel, 2016; Liu et al., 2020; Ming et al., 2022a).

### 5.3 Baselines

We compare the WOOD model against a broad range of CLIP-based OOD detection baselines, including zero-shot (1-2) and finetuned approaches (3-6), as well as the weakly-supervised implementation (7-8) of previous detection methods based on auxiliary OOD data.

1. **MCM-OOD** (Ming et al., 2022a). This method uses zero-shot CLIP for multi-modal OOD detection based on Maximum Concept Matching (MCM). It can only detect visual OODs, i.e. OOD images, in one type of scenario in which a given image is not aligned with its label in the training data.
2. **GL-MCM-OOD** (Miyai et al., 2023). As an extension of MCM-OOD, this work leverages both local and global visual-text alignments of CLIP features, known as Global-Local MCM (GL-MCM), to detect OOD samples containing ID classes with OOD features in the background in the zero-shot manner. However, it still only detects visual OODs.
3. **CoOp-MCM** (Miyai et al., 2023). This method adopts prompt learning to finetune the pre-trained CLIP using a set of learnable context vectors (Context Optimization). In addition, it uses MCM scores to detect visual OODs.
4. **NPOS-MCM** (Tao et al., 2023). This method finetunes the pretrained CLIP for enhanced OOD detection by generating artificial outliers from the boundary ID data selected based on the non-parametric k-nearest-neighbor (k-NN) distance. It also utilizes MCM scores to detect visual OODs.
5. **CLIP-N** (Wang et al., 2023b). This model adopts a learnable "no" text encoder in addition to the pre-trained CLIP to capture negation semantics. It also uses a threshold-free inference algorithm to perform OOD detection by utilizing the association between negation semantics from "no" prompts and the image encoder.
6. **LoCoOp** (Miyai et al., 2024). This work extends CoOp by performing OOD regularization that utilizes the CLIP local features as OOD features during training, since local features tend to have OOD related nuances such as background objects. It adopts GL-MCM scores to detect visual OODs.
7. **CLIP-Energy** (Liu et al., 2020). This approach adopts auxiliary OOD data to finetune the CLIP model using energy scores. For a fair comparison, we utilize the same number of OOD samples as our method for energy finetuning.
8. **CLIP-BCE** (Liznerski et al., 2022). This model finetunes the pre-trained CLIP with a binary classifier (BCE) in the supervised manner using our OOD samples as outliers for Outlier Exposure (Hendrycks et al., 2018), which aims to maximize the similarity of an image and the ID label set and minimize that for OOD samples.

### 5.4 Main Results

We conduct extensive experiments to thoroughly evaluate the detection performance of WOOD using three benchmark datasets. Here, recall that the proposed WOOD is weakly-supervised on training datasets containing 97% ID samples and 1% generated OOD samples for each of the three scenarios. To create a balanced evaluation setting, the test datasets are equally distributed for ID and our three OOD scenarios (25% each). We report the results on three subsets of the test dataset for each scenario (i.e. Misaligned+ID,

New-domain+ID, and Noisy+ID), as well as the overall performance on the entire test set. Furthermore, we conduct experiments to compare the computational cost of WOOD with that of the baselines in Appendix C.

Table 2: Performance comparison of different methods for OOD detection on CUB-200 dataset averaged over three random seeds. The standard deviations across random seeds are reported following $\pm$. Higher AUROC and lower FPR95 indicate better performance.

| Methods | Misaligned+ID | | New-domain+ID | | Noisy+ID | | All Scenarios+ID | |
|---|---|---|---|---|---|---|---|---|
| | AUROC↑ | FPR95↓ | AUROC↑ | FPR95↓ | AUROC↑ | FPR95↓ | AUROC↑ | FPR95↓ |
| Zero-shot | | | | | | | | |
| MCM-OOD | 47.16 ± 2.74 | 94.76 ± 1.26 | 81.42 ± 0.17 | 62.72 ± 1.01 | 51.62 ± 2.68 | 93.82 ± 5.79 | 60.07 ± 0.41 | 90.52 ± 1.29 |
| GL-MCM-OOD | 46.90 ± 1.37 | 96.33 ± 0.38 | 79.88 ± 1.07 | 72.50 ± 1.09 | 49.60 ± 3.91 | 96.25 ± 0.90 | 58.80 ± 2.11 | 94.42 ± 1.53 |
| Finetuned | | | | | | | | |
| CoOp-MCM | 48.61 ± 1.49 | 95.76 ± 0.67 | 90.30 ± 0.90 | 48.83 ± 2.02 | 52.83 ± 3.30 | 95.55 ± 0.68 | 63.91 ± 1.88 | 94.11 ± 0.77 |
| NPOS-MCM | 49.46 ± 1.76 | 94.58 ± 1.51 | 52.34 ± 2.69 | 92.67 ± 1.81 | 46.71 ± 0.64 | 93.83 ± 1.01 | 49.50 ± 0.96 | 94.08 ± 1.38 |
| CLIP-N | 49.96 ± 2.36 | 96.00 ± 1.80 | 80.21 ± 4.86 | 83.33 ± 9.75 | 49.64 ± 0.64 | 95.17 ± 1.51 | 59.94 ± 1.74 | 95.17 ± 2.75 |
| LoCoOp | 49.25 ± 2.68 | 95.92 ± 1.13 | 91.72 ± 1.66 | 42.58 ± 3.59 | 54.20 ± 3.66 | 92.92 ± 0.14 | 65.05 ± 2.62 | 91.90 ± 0.98 |
| Weakly-supervised | | | | | | | | |
| CLIP-Energy | 49.45 ± 1.43 | 95.25 ± 2.05 | 78.51 ± 1.54 | 83.50 ± 4.11 | 50.08 ± 4.36 | 95.67 ± 0.63 | 59.35 ± 2.81 | 93.58 ± 0.72 |
| CLIP-BCE | 50.37 ± 3.25 | 94.83 ± 0.29 | 88.61 ± 3.16 | 29.50 ± 5.25 | 95.10 ± 0.39 | 15.67 ± 3.13 | 78.08 ± 1.87 | 46.87 ± 2.49 |
| Ours | **91.71 ± 0.53** | **28.47 ± 1.25** | **99.08 ± 0.92** | **3.63 ± 1.18** | **96.73 ± 0.86** | **6.83 ± 1.26** | **95.18 ± 0.25** | **13.98 ± 0.60** |

Table 3: Performance comparison of different methods for OOD detection on MIMIC-CXR dataset averaged over three random seeds. The standard deviations across random seeds are reported following $\pm$. Higher AUROC and lower FPR95 indicate better performance.

| Methods | Misaligned+ID | | New-domain+ID | | Noisy+ID | | All Scenarios+ID | |
|---|---|---|---|---|---|---|---|---|
| | AUROC↑ | FPR95↓ | AUROC↑ | FPR95↓ | AUROC↑ | FPR95↓ | AUROC↑ | FPR95↓ |
| Zero-shot | | | | | | | | |
| MCM-OOD | 53.43 ± 1.20 | 96.89 ± 0.50 | 50.92 ± 1.61 | 93.54 ± 0.32 | 47.19 ± 2.17 | 96.01 ± 1.70 | 50.49 ± 1.46 | 95.55 ± 1.18 |
| GL-MCM-OOD | 54.15 ± 0.43 | 95.54 ± 0.53 | 57.05 ± 1.00 | 90.05 ± 2.14 | 47.15 ± 1.58 | 96.29 ± 0.95 | 52.78 ± 0.95 | 94.41 ± 0.67 |
| Finetuned | | | | | | | | |
| CoOp-MCM | 33.41 ± 1.51 | 97.90 ± 0.78 | 66.82 ± 0.92 | 69.46 ± 0.54 | 50.76 ± 1.02 | 94.80 ± 0.88 | 53.00 ± 1.15 | 95.27 ± 1.02 |
| NPOS-MCM | 47.83 ± 0.54 | 94.90 ± 1.55 | 57.81 ± 0.49 | 88.64 ± 1.62 | 52.22 ± 0.72 | 94.94 ± 1.57 | 52.62 ± 0.20 | 93.19 ± 1.28 |
| CLIP-N | 64.21 ± 0.93 | 88.58 ± 2.34 | 50.69 ± 3.98 | 86.56 ± 3.39 | 50.93 ± 1.07 | 95.16 ± 2.05 | 55.28 ± 1.72 | 91.24 ± 0.77 |
| LoCoOp | 33.73 ± 0.78 | 97.34 ± 0.45 | 62.27 ± 0.88 | 68.44 ± 2.98 | 51.11 ± 0.60 | 94.12 ± 1.04 | 49.04 ± 0.30 | 94.89 ± 0.81 |
| Weakly-supervised | | | | | | | | |
| CLIP-Energy | 64.71 ± 2.56 | 89.80 ± 2.64 | 93.32 ± 1.99 | 54.51 ± 9.14 | 50.05 ± 0.44 | 94.12 ± 1.93 | 69.33 ± 1.69 | 89.94 ± 2.60 |
| CLIP-BCE | 56.52 ± 3.35 | 93.24 ± 1.52 | 99.21 ± 1.15 | 1.16 ± 1.08 | 96.37 ± 2.31 | 11.19 ± 7.09 | 85.45 ± 1.66 | 32.70 ± 0.97 |
| Ours | **89.02 ± 0.67** | **37.66 ± 0.85** | **99.81 ± 0.25** | **0.47 ± 0.63** | **96.95 ± 0.55** | **8.03 ± 1.15** | **94.48 ± 0.50** | **15.39 ± 0.31** |

Table 4: Performance comparison of different methods for OOD detection on COCO dataset averaged over three random seeds. The standard deviations across random seeds are reported following $\pm$. Higher AUROC and lower FPR95 indicate better performance.

| Methods | Misaligned+ID | | New-domain+ID | | Noisy+ID | | All Scenarios+ID | |
|---|---|---|---|---|---|---|---|---|
| | AUROC↑ | FPR95↓ | AUROC↑ | FPR95↓ | AUROC↑ | FPR95↓ | AUROC↑ | FPR95↓ |
| Zero-shot | | | | | | | | |
| MCM-OOD | 40.86 ± 2.31 | 97.99 ± 0.53 | 71.15 ± 1.44 | 68.05 ± 1.24 | 47.39 ± 1.75 | 96.15 ± 0.62 | 53.13 ± 1.71 | 95.42 ± 1.04 |
| GL-MCM-OOD | 35.44 ± 4.01 | 99.73 ± 0.46 | 86.29 ± 3.87 | 46.40 ± 16.86 | 47.66 ± 6.59 | 96.53 ± 1.67 | 56.50 ± 4.81 | 98.40 ± 1.31 |
| Finetuned | | | | | | | | |
| CoOp-MCM | 56.31 ± 1.33 | 95.50 ± 2.00 | 68.21 ± 0.72 | 72.16 ± 3.64 | 44.51 ± 1.94 | 96.13 ± 0.78 | 56.34 ± 1.11 | 93.45 ± 2.05 |
| NPOS-MCM | 34.06 ± 0.93 | 96.04 ± 0.74 | 74.81 ± 1.33 | 92.57 ± 1.17 | 49.32 ± 0.95 | 95.98 ± 0.71 | 52.73 ± 0.46 | 95.57 ± 0.73 |
| CLIP-N | 61.97 ± 1.66 | 85.24 ± 2.31 | 72.73 ± 1.45 | 78.61 ± 2.01 | 55.33 ± 0.43 | 92.14 ± 1.67 | 63.35 ± 0.96 | 87.91 ± 1.47 |
| LoCoOp | 55.11 ± 2.17 | 94.68 ± 1.53 | 89.56 ± 0.57 | 43.77 ± 3.92 | 43.74 ± 1.96 | 96.10 ± 0.79 | 62.80 ± 1.34 | 93.75 ± 0.83 |
| Weakly-supervised | | | | | | | | |
| CLIP-Energy | 57.83 ± 1.34 | 84.73 ± 4.18 | 96.07 ± 0.50 | 24.46 ± 9.46 | 52.65 ± 1.84 | 93.84 ± 2.18 | 68.85 ± 1.16 | 86.88 ± 3.31 |
| CLIP-BCE | 56.95 ± 1.85 | 93.56 ± 2.40 | 98.39 ± 0.17 | 6.69 ± 0.38 | 98.73 ± 0.42 | 6.09 ± 1.30 | 84.69 ± 0.77 | 35.47 ± 1.38 |
| Ours | **99.78 ± 0.07** | **1.14 ± 0.32** | **99.88 ± 0.06** | **0.88 ± 0.19** | **99.08 ± 0.21** | **2.89 ± 0.35** | **99.58 ± 0.07** | **1.54 ± 0.31** |

We first assess the performance of the proposed WOOD on CUB-200 dataset. Table 2 illustrates the comparison results of different OOD detection methods using three random seeds. We can observe from this table that our method is effective in all three OOD scenarios, and its overall performance significantly exceeds the baselines. The main reason is that our method leverages both contrastive learning and a binary classifier for OOD detection on all three scenarios, as motivated by Fig. 2. Evidently, zero-shot methods such as MCM-OOD and GL-MCM-OOD do not perform well since they are only designed for detecting visual OODs by querying an image to check whether the returned features from CLIP align with training labels. In addition, although finetuned CLIP methods are designed to improve the detection performance on *new-domain* OODs,

we discover that finetuning approaches such as NPOS-MCM and CLIP-N can have an adverse effect on performance when when there are not many OOD training images from classes distinguishable from "bird". In particular, when NPOS-MCM generates training OOD examples on the k-NN boundary of ID images, these examples can overlap with ID images from another classes instead of OODs from COCO-bird, resulting in worse performance than zero-shot methods. Also, while CLIP-N can embed negation semantics from images to "no" text prompts, these semantics may not be sufficiently distinct from ID "bird" features. On the other hand, CoOp and LoCoOp effectively build contexts from CLIP image features around the associated class labels, resulting in better *new-domain* detection. Despite taking advantage of OOD training data, the small number of training OODs is not sufficient for CLIP-Energy and CLIP-BCE to beat context prompt finetuning for *new-domain* examples. Similarly to our method, CLIP-BCE can utilize weakly-supervised binary classification to effectively detect *noisy* visual OOD examples, but it still fails to detect *misaligned* textual OODs. In contrast, our method can detect both visual and textual OODs. Still, it is very challenging to identify a pair of OOD samples when an image is slightly unaligned with its textual information. Specifically in our experiment, *misaligned* test samples include many similar images as those in the ID samples, while the textual descriptions also contain "bird" with hardly different fine-grained features. As a result, the FPR95 score for detecting *misaligned* OODs is higher compared to other scenarios.

Next, we also show that WOOD can identify OOD samples in multi-modal medical data, MIMIC-CXR. As illustrated in Table 3, it can be observed that the proposed method effectively detects OODs in all three OOD scenarios and its overall performance is better than the baselines. Here, note that similar to CUB-200, the FPR95 of the *misaligned* scenario is also high since the test images have the same class label "chest scans" as those in the training data and only have *slightly different textual descriptions*. Furthermore, zero-shot and finetuned CLIP methods fail to detect OODs in all three scenarios because the pre-trained CLIP lacks the alignment capability for the medical image features and data labels, which is absent from its pretraining data. With additional weakly-supervised training, CLIP-Energy and CLIP-BCE can learn better energy-based OOD scores or binary predictions respectively to significantly improve the detection performance for *new-domain* samples. However, CLIP-BCE falls short in detecting *noisy* OOD samples in this experiment with higher false positives. One possible explanation is that while a classifier is effective at detecting visual noise, the binary classifier in CLIP-BCE aligns the image features with the collection of ID class labels, instead of the corresponding textual descriptions using contrastive learning. On the other hand, this issue is clearly mitigated by the interplay of our two components where our proposed method first aligns medical images with their detailed textual description with constrastive learning, then combines the learned features in conjunction with feature sparsity to make accurate predictions for all three OOD scenarios simultaneously.

Furthermore, we apply the proposed WOOD to detect OOD samples on COCO dataset. We compare the detection performance of different methods as shown in Table 4. It can be seen that our approach is able to effectively detect OODs in all three OOD scenarios while the baseline methods can only detect *new-domain* or *noisy* OODs. In addition, since COCO and its corresponding *new-domain* dataset GCC both mostly contain images with the class label "person", zero-shot and finetuned CLIP baselines with except GL-MCM-OOD and LoCoOp struggle to detect these OODs. The reason why GL-MCM-OOD and LoCoOp outperform the other baselines is that they can capture local CLIP features representative of OOD in the image background in the learnable context prompts around the class labels, as previously demonstrated by Miyai et al. (2023) and Miyai et al. (2024). However, by utilizing a small number of OOD training data, weakly-supervised classification methods such as CLIP-BCE and ours can capture these OOD nuances even more effectively, as evidenced by their superior performance for detecting *new-domain* OODs, as well as *noisy* visual OOD examples. Moreover, the detection performance for *misaligned* OODs is much higher than the previous two datasets, since COCO has a much larger training set that contains many easily distinguishable classes (e.g. person, cat, bird, etc.).

## 5.5   Ablation Studies

In this section, we conduct ablation studies to investigate the effect of some important components, the weight in the objective function, as well as the Hinge loss and its hyperparameter on the detection performance.

Table 5: Impact of ablating the binary classifier (BC) and the contrastive learning (CL) components from the proposed WOOD method. We report the results averaged over three random seeds. It can be seen that our proposed method outperforms both its counterparts without the binary classifier or the contrastive learning module for detecting three OOD scenarios on all three datasets.

| Models | Misaligned+ID | | New-domain+ID | | Noisy+ID | | All Scenarios+ID | |
|---|---|---|---|---|---|---|---|---|
| | AUROC↑ | FPR95↓ | AUROC↑ | FPR95↓ | AUROC↑ | FPR95↓ | AUROC↑ | FPR95↓ |
| CUB-200 | | | | | | | | |
| WOOD w/o BC | 90.84 | 31.67 | 86.94 | 40.08 | 51.19 | 94.67 | 76.32 | 55.47 |
| WOOD w/o CL | 49.26 | 93.25 | 97.62 | 7.25 | 88.74 | 33.83 | 78.52 | 44.72 |
| WOOD | **91.71** | **28.47** | **99.08** | **3.63** | **96.73** | **6.83** | **95.18** | **13.98** |
| MIMIC-CXR | | | | | | | | |
| WOOD w/o BC | 86.75 | 51.52 | 86.75 | 48.50 | 51.52 | 94.48 | 72.58 | 68.43 |
| WOOD w/o CL | 68.34 | 85.91 | 99.32 | 3.39 | 51.21 | 95.00 | 73.15 | 61.42 |
| WOOD | **89.02** | **37.66** | **99.81** | **0.47** | **96.95** | **8.03** | **94.48** | **15.39** |
| COCO | | | | | | | | |
| WOOD w/o BC | 99.73 | 1.60 | 90.43 | 27.12 | 53.59 | 94.02 | 81.25 | 41.24 |
| WOOD w/o CL | 77.90 | 80.91 | 99.48 | 2.02 | 96.80 | 10.22 | 91.39 | 31.05 |
| WOOD | **99.78** | **1.14** | **99.88** | **0.88** | **99.08** | **2.89** | **99.58** | **1.54** |

**Effect of Main Model Components.** First, to assess the efficacy of distinct components within the proposed WOOD method, we conduct an ablation study by systematically excluding individual components and measuring the OOD detection performance. The results are presented in Table 5. From this table, we can draw two conclusions: (i) notably, our method incorporating both the binary classifier (BC) and the contrastive learning (CL) components demonstrates superior performance; and (ii) each component exhibits a notable influence on performance, affirming the necessity of their inclusion in our method. More specifically, the WOOD model without the binary classifier can detect *misaligned* OODs well, but its detection accuracy suffers when the OOD features appear outside the scope of the textual descriptions (i.e., in the background) for the *new-domain* and *noisy* scenarios. On the other hand, the WOOD model without contrastive learning lacks knowledge of the alignment between images and their corresponding textual descriptions, thus failing to detect *misaligned* OODs. Therefore, our design of the proposed WOOD allows these two components to complement each other and significantly enhance the OOD detection performance for three scenarios simultaneously.

**Effect of Weight in the Objective.** We study the effect of the weight in the objective function ($\lambda$ in Eq. (7)) on the performance of OOD detection. We can see from Table 6 that the proposed WOOD model has the best performance when $\lambda = 0.4$ as $\lambda$ decreases from 0.5 to 0.2 on all three datasets. Note that in the previous ablation study (Table 5), we demonstrate the detection performance when our method utilizes only the contrastive learning module when $\lambda = 0$ in Eq. 7, or only the binary classifier when $\lambda = 1$. Additionally, when $\lambda = 0.5$, the contrastive learning module and the binary classifier have equal weights in the objective. As illustrated in Table 6, for more challenging datasets with highly similar images such as CUB-200 and MIMIC-CXR, low $\lambda$ values have much more pronounced negative impact on the OOD detection performance, as the model becomes less effective in detecting *new-domain* and *noisy* scenarios. This further supports our motivation for combining two different modules in the proposed WOOD.

Table 6: Impact of the weight of the binary classifier in the objective on detection performance. We report the results averaged over three random seeds. It can be seen that the proposed WOOD performs very well when $\lambda = 0.4$ on all three datasets.

| $\lambda$ | Dataset | | | | | |
|---|---|---|---|---|---|---|
| | CUB-200 | | MIMIC-CXR | | COCO | |
| | AUROC↑ | FPR95↓ | AUROC↑ | FPR95↓ | AUROC↑ | FPR95↓ |
| 0.5 | 95.15 | 15.72 | 94.41 | 17.20 | 99.49 | 1.66 |
| 0.4 | **95.18** | **13.98** | **94.48** | **15.39** | **99.58** | 1.54 |
| 0.3 | 90.55 | 30.20 | 94.08 | 19.33 | 99.46 | **1.20** |
| 0.2 | 82.57 | 40.97 | 85.28 | 38.11 | 99.39 | 1.56 |

**Effect of Hinge Loss and its Hyperparameter.** We also explore the impact of the margin hyperparameter $m$ used in the Hinge loss (in Eq. 2 and Eq. 3) on the detection performance. Table 7 illustrates

the detection performance under different margin parameters. In addition, we ablate the Hinge loss (Eq. 4) to investigate its impact on the model performance. Specifically, the model without the Hinge loss utilizes $\mathcal{L}_1$ as defined in Eq. 1 for contrastive learning. We can observe from Table 7 that our method performs the best when $m = 0.2$ for CUB-200 and MIMIC-CXR. Since COCO has many easily distinguishable classes, our model benefits from using a slightly large margin $m = 0.3$. Moreover, our detection method without the Hinge loss ($\mathcal{L}_1$) does not perform well for CUB-200 and MIMIC-CXR. Therefore, we can conclude that this Hinge loss plays an important role in multi-modal OOD detection.

Table 7: Impact of Hinge loss in the objective on detection performance. We report the averaged results from three random seeds. It can be observed that when $m = 0.2$, the proposed model has the best performance on CUB-200 and MIMIC-CXR datasets. For COCO data, it performs best as $m = 0.3$.

| $\mathcal{L}_{cl}$ | Dataset | | | | | |
| | CUB-200 | | MIMIC-CXR | | COCO | |
| | AUROC↑ | FPR95↓ | AUROC↑ | FPR95↓ | AUROC↑ | FPR95↓ |
|---|---|---|---|---|---|---|
| No Hinge | 81.42 | 41.11 | 86.90 | 29.78 | 99.01 | 2.09 |
| $m = 0.1$ | 94.27 | 18.95 | 92.74 | 21.84 | 99.35 | 1.87 |
| $m = 0.2$ | **95.18** | **13.98** | **94.48** | **15.39** | 99.48 | 1.65 |
| $m = 0.3$ | 94.68 | 17.41 | 93.21 | 20.75 | **99.58** | **1.54** |
| $m = 0.4$ | 92.57 | 25.61 | 93.77 | 19.80 | 99.37 | 1.87 |

**Effect of the Feature Sparsity Regularizer.** Last but not least, we conduct ablations of the Feature Sparsity Regularizer (FSR) used in the binary classifier to measure its usefulness for OOD detection. From Table 8, it can be observed that using sparse features for the binary classifier is beneficial for the detection performance, especially for the medical data. Since the textual description for each image contains about 40 words on average in MIMIC-CXR, the FSR module encourages the binary classifier to gather highly informative features from the text modality for OOD classification.

Table 8: Impact of the Feature Sparsity Regularizer (FSR) used in the binary classifier on detection performance. Averaged results from three random seeds show that employing FSR leads to superior performance across all three datasets.

| FSR | Dataset | | | | | |
| | CUB-200 | | MIMIC-CXR | | COCO | |
| | AUROC↑ | FPR95↓ | AUROC↑ | FPR95↓ | AUROC↑ | FPR95↓ |
|---|---|---|---|---|---|---|
| No | 94.75 | 15.64 | 84.41 | 37.21 | 99.23 | 1.64 |
| Yes | **95.18** | **13.98** | **94.48** | **15.39** | **99.58** | **1.54** |

# 6 Conclusions and Future Work

In this paper, we developed WOOD, a general-purpose multi-modal OOD detection framework that integrates contrastive learning and a binary classifier in a weakly-supervised manner. Concretely, we employed Hinge loss in the contrastive learning component to amplify the dissimilarity between ID and OOD pairs. In addition, we developed a Feature Sparsity Regularizer to selectively incorporate crucial features from data of two modalities for the binary classifier. We also devised a novel scoring metric to effectively fuse predictions from both components. The evaluation results validated that the proposed hybrid approach excels in identifying anomalies across three different OOD scenarios. Our future work will focus on applying the proposed WOOD to more OOD scenarios to better represent the wide spectrum of OOD environments in the real world.

# Acknowledgements

Research reported in this paper was sponsored in part by NSF under award CPS NSF-2311086 and Faculty Research Grant at William & Mary 141446.

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

## A  Additional Related Work on Multi-Modal OOD Generalization

Besides OOD detection, OOD generalization is also an emerging topic of machine learning research that focuses on complex scenarios where the testing distribution is different from those of the training data (Liu et al., 2021). In particular, multi-modal OOD generalization aims to improve the robustness of multi-modal models on out-of-distribution testing data (Shu et al., 2023; Tu et al., 2023; Jung et al., 2023; Huang et al., 2023; Zhang et al., 2024). Shu et al. (2023) developed a fine-tuning approach using class adaptive margins to adapt CLIP models to OOD scenarios where both domain shifts and open classes may occur on the unseen testing data. Tu et al. (2023) studied the robustness of CLIP to variations in fine-grained visual factors such as shapes and textures, and adopted post-hoc prediction uncertainty calibration for better classification accuracy across diverse distributions. Jung et al. (2023) considered various types of noise as OOD multi-modal inputs, and proposed Multi-modal Neural Processes to generalize neural processes for multi-modal classification and uncertainty estimation. Moreover, Huang et al. (2023) focused on generalizing Multi-modal Large Language Models (MLLMs) for out-of-domain vision tasks, utilizing in-context learning to detect and rectify incorrect model predictions. Similarly to Huang et al. (2023), Zhang et al. (2024) investigated the generalization ability of MLLMs in out-of-domain OOD scenarios and studied the effectiveness of in-context learning in bolstering MLLMs' adaptability.

In summary, these studies primarily addressed the generalization of multi-modal models for specific types of OOD, such as out-of-domain or noisy classes, where OOD detection took the back seat as a secondary objective. In contrast, our study focused on enhancing the detection ability of multi-modal models for OOD samples belonging to several scenarios including misalignment, out-of-domain, and noise simultaneously in the fine-grained manner.

## B  Evaluation of CLIP's Generalization to Noisy OOD Samples

In this section, we evaluate the CLIP model (ViT-B/16) using linear probing for bird species classification on the CUB-200 dataset. Then, we experimentally add Gaussian blur to different percentages of image patches, ranging from 10% up to 50%, and report the Macro-F1 scores to measure the model's classification accuracy for noisy images, as shown in Table 9 below. In particular, even when little noise is added (10% of patches), the prediction performance of the CLIP model drops by 6% if half of the testing images contain noise, and by 14% if all images are noisy. Additionally, the CLIP model only achieves a 60.12% Macro-F1 score on the noisy testing dataset with 50% noisy patches, which is a 22% reduction in performance. Therefore, we conclude that CLIP is not robust to noisy images.

Table 9: The OOD generalization evaluation for classification of noisy samples on the CUB-200 dataset. The performance is measured using Macro-F1 scores (%) averaged over three random seeds.

| % Samples \ % Patches | 0 | 10 | 25 | 50 |
|---|---|---|---|---|
| 0 | **77.62** | — | — | — |
| 25 | — | 75.62 | 74.43 | 73.58 |
| 50 | — | 72.78 | 70.04 | 69.22 |
| 100 | — | 66.74 | 62.40 | 60.12 |

## C  Evaluation of Computational Cost

In this section, we conduct experiments to analyze the computational costs of our proposed method, and other baselines which utilize fine-tuning for CLIP. Additionally, we develop a lightweight variant of our approach, enabling training solely on the final two transformers layers of the CLIP image and text encoders. The computational cost results are outlined in Table 10, while Table 11 showcases a comparative evaluation of OOD detection performance between the lightweight and fully-trained versions across three datasets. Notably, our lightweight method demonstrates competitive training speed compared to the baselines while improving OOD detection performance. Moreover, partially training the CLIP model achieves competitive OOD detection performance compared to both the best baseline, CLIP-BCE, and its fully-trained counterpart

on the CUB-200 and COCO datasets, as shown in Table 11. However, the lightweight approach does not perform well on the MIMIC-CXR dataset due to CLIP's lack of pretraining on medical data.

Table 10: Computational cost comparison between our proposed method and the baselines that finetune CLIP. The training speed is measured by training throughput in samples/second averaged over three random seeds. It can be observed that our lightweight method demonstrates competitive training speed compared to the baselines while improving OOD detection performance. Additionally, we specify whether each model component such as Image Encoder, Text Encoder, and Classifier is fully-trained (●), partially-trained (◑), frozen (○), or absent (_).

| Methods | Image Encoder | Text Encoder | Classifier | Throughput (samples/s) | | |
|---|---|---|---|---|---|---|
| | | | | CUB200 | MIMIC-CXR | COCO |
| CoOp-MCM | ○ | ● | _ | 107.0 ± 0.19 | 123.3 ± 0.27 | 139.9 ± 0.43 |
| NPOS-MCM | ◑ | ○ | _ | 123.3 ± 0.36 | 108.7 ± 0.44 | 125.1 ± 0.57 |
| CLIP-N | ○ | ● | _ | 99.7 ± 0.28 | 90.5 ± 0.37 | 102.7 ± 0.49 |
| LoCoOp | ○ | ● | _ | 96.3 ± 0.17 | 110.0 ± 0.24 | 125.9 ± 0.39 |
| CLIP-Energy | ● | ● | ● | 67.3 ± 0.24 | 62.8 ± 0.30 | 69.0 ± 0.46 |
| CLIP-BCE | ● | ● | ● | 80.4 ± 0.27 | 73.5 ± 0.37 | 83.0 ± 0.45 |
| Ours (light) | ◑ | ◑ | ● | 110.3 ± 0.17 | 105.1 ± 0.28 | 115.4 ± 0.38 |
| Ours (full) | ● | ● | ● | 71.7 ± 0.12 | 66.0 ± 0.25 | 73.1 ± 0.34 |

Table 11: Performance comparison between the lightweight and fully-trained version of our proposed method averaged over three random seeds. It can be seen that partially training the CLIP encoders results in marginal performance reduction on the CUB-200 and COCO datasets, but full training is necessary for MIMIC-CXR due to the lack of CLIP pretraining on medical data.

| Components | Misaligned+ID | | New-domain+ID | | Noisy+ID | | All Scenarios+ID | |
|---|---|---|---|---|---|---|---|---|
| | AUROC↑ | FPR95↓ | AUROC↑ | FPR95↓ | AUROC↑ | FPR95↓ | AUROC↑ | FPR95↓ |
| CUB-200 | | | | | | | | |
| CLIP-BCE | 50.37 ± 3.25 | 94.83 ± 0.29 | 88.61 ± 3.16 | 29.50 ± 5.25 | 95.10 ± 0.39 | 15.67 ± 3.13 | 78.08 ± 1.87 | 46.87 ± 2.49 |
| Ours (light) | 89.84 ± 1.60 | 36.67 ± 3.82 | 98.90 ± 0.81 | 4.33 ± 3.17 | 90.74 ± 1.69 | 21.25 ± 3.54 | 93.39 ± 0.33 | 19.86 ± 0.40 |
| Ours (full) | **91.71 ± 0.53** | **28.47 ± 1.25** | **99.08 ± 0.92** | **3.63 ± 1.18** | **96.73 ± 0.86** | **6.83 ± 1.26** | **95.18 ± 0.25** | **13.98 ± 0.60** |
| MIMIC-CXR | | | | | | | | |
| CLIP-BCE | 56.52 ± 3.35 | 93.24 ± 1.52 | 99.21 ± 1.15 | 1.16 ± 1.08 | 96.37 ± 2.31 | 11.19 ± 7.09 | 85.45 ± 1.66 | 32.70 ± 0.97 |
| Ours (light) | 78.37 ± 1.82 | 63.23 ± 1.60 | 99.69 ± 0.12 | 1.28 ± 0.48 | 51.72 ± 0.87 | 93.97 ± 1.88 | 76.59 ± 0.79 | 52.82 ± 1.10 |
| Ours (full) | **89.02 ± 0.67** | **37.66 ± 0.85** | **99.81 ± 0.25** | **0.47 ± 0.63** | **96.95 ± 0.55** | **8.03 ± 1.15** | **94.48 ± 0.50** | **15.39 ± 0.31** |
| COCO | | | | | | | | |
| CLIP-BCE | 56.95 ± 1.85 | 93.56 ± 2.40 | 98.39 ± 0.17 | 6.69 ± 0.38 | 98.73 ± 0.42 | 6.09 ± 1.30 | 84.69 ± 0.77 | 35.47 ± 1.38 |
| Ours (light) | 99.55 ± 0.23 | 1.60 ± 0.39 | 99.64 ± 0.12 | 1.26 ± 0.29 | 93.58 ± 0.07 | 15.43 ± 0.75 | 97.68 ± 0.08 | 5.75 ± 0.28 |
| Ours (full) | **99.78 ± 0.07** | **1.14 ± 0.32** | **99.88 ± 0.06** | **0.88 ± 0.19** | **99.08 ± 0.21** | **2.89 ± 0.35** | **99.58 ± 0.07** | **1.54 ± 0.31** |

