# OpenReview forum: "A General-Purpose Multi-Modal OOD Detection Framework"
_TMLR — Accepted by TMLR_

### Review · Reviewer_wSrD · 2024-03-29

**Summary Of Contributions:**

This paper studies OOD detection using multi-modal models such as CLIP. There are three types of data that need to be detected in multi-modal OOD detection, namely ID misaligned data, OOD aligned data, and OOD misaligned data. In order to address this problem, the CLIP models need to be trained to identify both the misalignment between image and text and the image from new domains. In this paper, the authors propose to conduct contrastive learning to minimize the confidence in OOD data meanwhile increasing the confidence in ID data. Instead of using typical contrastive loss, the authors propose to use Hinge loss to constrain the similarity score. Moreover, for the detection of the OOD data, the authors propose a binary classifier to classify OOD data from ID data.

**Audience:**

Yes

**Broader Impact Concerns:**

No ethic issue.

**Claims And Evidence:**

Yes

**Requested Changes:**

Please see weaknesses for details.

**Strengths And Weaknesses:**

Strengths:
- This paper studies OOD detection using multi-modal models, which is quite a novel and interesting study direction.
- The writing is good, is not hard to understand the intention of the authors.
- The Experimental improvement is very good and significantly surpasses many baseline methods.

Weaknesses:
- The proposed method needs clearer motivation to justify its design. Why using Hinge loss instead of typical contrastive loss is not sufficiently and clearly explained. Why can Hinge loss constrain the similarity score? Why is a feature sparsity regularizer employed? Why can sparsity help OOD detection in this scenario?
The authors are referred to the following literature for OOD data under sparsity:
[1] Zhang et al., Can Subnetwork Structure be the Key to Out-of-Distribution Generalization?

- Training CLIP is computationally expensive. However, the training efficiency is not discussed. Compared to other baseline methods, the effectiveness has been largely improved, but the computational cost compared to other methods should be more elaboratively demonstrated.
- There are many loss terms and designs in the proposed approach. However, only contrastive learning and binary classification have been considered in the ablation study. There should be more components in this method, including using Hinge loss, sparsity terms, etc.
- Is this framework only applicable to CLIP? Are there any other models that can be used for this method?
- Missing several references about multi-modal models under OOD scenario:
[2] Jung et al., Beyond Unimodal: Generalising Neural Processes for Multimodal Uncertainty Estimation.
[2] Huang et al., Machine Vision Therapy: Multimodal Large Language Models Can Enhance Visual Robustness via Denoising In-Context Learning.
[3] Zhang et al., On the Out-Of-Distribution Generalization of Multimodal Large Language Models.

---

> ### Author Response · Authors · 2024-05-03
> **Response to Reviewer wSrD (Part 1/2)**
>
> Q1: The proposed method needs clearer motivation to justify its design. Why using Hinge loss instead of typical contrastive loss is not sufficiently and clearly explained. Why can Hinge loss constrain the similarity score? Why is a feature sparsity regularizer employed? Why can sparsity help OOD detection in this scenario?
> --
> **A1:** The motivation behind incorporating the Hinge loss is that we can explicitly regulate the learning process of the CLIP model for vision-language alignment and OOD separation simultaneously using the margin hyperparameter. Specifically, as we utilized a small number of _K_ OOD samples during training, the objective in Eq. 2 aims to encourage the cosine similarity difference between the aligned _N-K_ ID pairs and incorrect pairings to be larger than a margin _m_. Additionally, the objective in Eq. 3 ensures that  _K_ OOD images align with neither of the _N−K_ ID texts, nor its _K_ corresponding texts, by constraining their similarities under the margin. In contrast, the regular contrastive loss in Eq. 1 minimizes the cross entropy of cosine similarities between _N−K_ pairs and maximizes that of _K_ OOD pairs separately, without considering the relationship between them, leading to less effective OOD detection performance as demonstrated in our experiment.
>
> Previous works [1] [2] have demonstrated that DNNs are prone to overfitting to observed patterns in the training data, resulting in activations of irrelevant features from OOD inputs and impaired OOD detection capability. To mitigate this issue, these works proposed the utilization of feature sparsity. In line with this motivation, our method incorporates the Feature Sparsity Regularizer to promote sparse vision-language representations as inputs for the binary classifier. This strategy not only helps circumvent potentially over-parameterized representations from the CLIP model, but also facilitates effective fusion of relevant features from two modalities for OOD detection.
>
> [1] Zhang et al. Can Subnetwork Structure be the Key to Out-of-Distribution Generalization?. ICML 2021.
>
> [2] Sun et al. Dice: Leveraging sparsification for out-of-distribution detection. ECCV 2022.
>
> Q2: Training CLIP is computationally expensive. However, the training efficiency is not discussed. Compared to other baseline methods, the effectiveness has been largely improved, but the computational cost compared to other methods should be more elaboratively demonstrated.
> --
> **A2**: Following your suggestion, we have conducted experiments to analyze the computational costs of our proposed method and other baselines utilizing finetuning for CLIP. Additionally, we developed a lightweight variant of our approach, enabling training solely on the final two transformers layers of the CLIP image and text encoders. The computational cost results are outlined in Table 1, while Tables 2-4 showcase a comparative evaluation of OOD detection performance between the lightweight and fully-trained versions across three datasets. Notably, our lightweight method demonstrates competitive training speed compared to the baselines. Moreover, partially training the CLIP model yields competitive detection performance with the fully-trained version on the CUB-200 and COCO datasets, as illustrated in Tables 2 and 3, respectively. However, as observed in Table 4, the lightweight approach does not perform well on the MIMIC-CXR dataset due to CLIP's lack of pretraining on medical data. Consequently, we opted to report the OOD detection results using the fully-trained version of our proposed method in the submission.
>
>
> Table 1: Computational cost comparison between the proposed method and other baselines that finetune CLIP.
> |Training Throughput (samples/s)↑|CUB-200|MIMIC-CXR|COCO|
> |-|-|-|-|
> |CoOp-MCM|107|123|140|
> |NPOS-MCM|123|109|125|
> |CLIP-N|100|91|103|
> |LoCoOp|96|110|126|
> |CLIP-Energy|67|63|69|
> |CLIP-BCE|80|74|83|
> |Ours (full)|72|66|73|
> |Ours (light)|110|105|115|
>
>
> Table 2: Evaluation of the lightweight method on CUB-200.
> |CUB-200| Misaligned+ID||New-domain+ID||Noisy+ID||All Scenarios+ID||
> |-|-|-|-|-|-|-|-|-|
> || AUROC↑|FPR95↓|AUROC↑|FPR95↓|AUROC↑|FPR95↓|AUROC↑|FPR95↓|
> |Ours (full)|**91.71**|**28.47**|**99.08**|**3.63**|**96.73**|**6.83**  |**95.18**|**13.98**|
> |Ours (light)|89.84|36.67|98.90|4.33|90.74|21.25|93.39|19.86|
>
>
> Table 3: Evaluation of the lightweight method on COCO.
> |COCO|Misaligned+ID||New-domain+ID||Noisy+ID||All Scenarios+ID||
> |-|-|-|-|-|-|-|-|-|
> ||AUROC↑|FPR95↓|AUROC↑|FPR95↓|AUROC↑|FPR95↓|AUROC↑|FPR95↓|
> |Ours (full)|**99.78**|**1.14**|**99.88**|**0.88**|**99.08**| **2.89**|**99.58**|**1.54**|
> |Ours (light)|99.55|1.40|99.64|1.26|93.58|15.43|97.68|5.75|
>
>
> Table 4: Evaluation of the lightweight method on MIMIC-CXR.
> |MIMIC-CXR|Misaligned+ID||New-domain+ID||Noisy+ID||All Scenarios+ID||
> |-|-|-|-|-|-|-|-|-|
> |Ours (full)|**89.02**|**37.66**|**99.81**| **0.47**|**96.95**| **8.03**| **94.48**| **15.39**|
> |Ours (light)|78.37|63.23|99.69|1.28|51.72|93.97|76.59|52.82|

---

> ### Author Response · Authors · 2024-05-03
> **Response to Reviewer wSrD (Part 2/2)**
>
> Q3: There are many loss terms and designs in the proposed approach. However, only contrastive learning and binary classification have been considered in the ablation study. There should be more components in this method, including using Hinge loss, sparsity terms, etc.
> --
> **A3**: We have conducted the suggested ablation studies with respect to the Hinge loss and the Feature Sparsity Regularizer (FSR) in Tables 7 and 8 in our submission respectively. To help you better revisit the results of these ablation studies, we highlight the results in Tables 5 and 6 below. In particular, as observed in Table 5, the proposed Hinge loss outperforms the typical contrastive loss (L1 in Eq. 1 in the submission), and it achieves the highest detection accuracy when the margin _m_ is set to 0.2 for CUB-200 and MIMIC-CXR datasets, and 0.3 for COCO. Additionally, we demonstrated in Table 6 that when the FSR is not utilized and the sparsity terms in the binary classifier objective (Eq. 6) are omitted, the OOD detection performance diminishes especially on the medical dataset MIMIC-CXR.
>
> Table 5: Impact of Hinge loss in the objective on detection performance averaged over from three random seeds.
> | Constrastive Loss | CUB-200 |        | MIMIC-CXR |        | COCO |        |
> |-----------|---------------|--------|---------------|--------|----------|--------|
> |           |     AUROC↑    | FPR95↓ |     AUROC↑    | FPR95↓ |  AUROC↑  | FPR95↓ |      AUROC↑      | FPR95↓ |
> |No Hinge| 81.42| 41.11| 86.90 |29.78| 99.01| 2.09
> |Hinge (_m=0.2_)| **95.18** |**15.39**| **94.48**| **15.39**| 99.48 |1.65|
> |Hinge (_m=0.3_)| 94.68 |17.41 |93.21 |20.75 |**99.58**| **1.54**|
>
> Table 6: Impact of Feature Sparsity Regularizer (FSR) in the binary classifier on detection performance averaged over from three random seeds.
> | FSR | CUB-200 |        | COCO |        | MIMIC-CXR |        |
> |-----------|---------------|--------|---------------|--------|----------|--------|
> |           |     AUROC↑    | FPR95↓ |     AUROC↑    | FPR95↓ |  AUROC↑  | FPR95↓ |      AUROC↑      | FPR95↓ |
> |No FSR| 94.75 |15.64 |84.41 |37.21| 99.23 |1.64|
> |FSR |**95.18**| **13.98**| **94.48**| **15.39** |**99.58** |**1.54**
>
> Q4: Is this framework only applicable to CLIP? Are there any other models that can be used for this method?
> --
> **A4**: Our proposed method focused on improving the OOD detection performance of vision-language models like CLIP utilizing contrastive learning for multimodal alignment. Given many emerging vision-language models [3][4][5] following CLIP, our method is applicable to these models for detecting vision-language OODs. For future work, we will extend this approach to other multimodal models with strong alignment capability between a wider range of modalities such as [6].
>
> [3] Li, Junnan, et al. Blip: Bootstrapping language-image pre-training for unified vision-language understanding and generation. PMLR, 2022.
>
> [4] Yu et al. CoCa: Contrastive Captioners are Image-Text Foundation Models. PMLR, 2022.
>
> [5] Alayrac et al. Flamingo: a visual language model for few-shot learning. NeuRIPS, 2022.
>
> [6] Girdhar et al. IMAGEBIND: One Embedding Space To Bind Them All. CVPR, 2023.
>
> Q5: Missing several references about multi-modal models under OOD scenario.
> --
> **A5**: We have discussed the suggested works as follows. [7] proposed Multimodal Neural Processes to generalize neural processes for multimodal classification and uncertainty estimation, however, their approach considered various types of noise as multimodal inputs, and focused solely on detecting noisy OOD samples. Moreover, [8] focused on generalizing Multi-modal Large Language Models (MLLMs) for out-of-domain vision tasks, utilizing in-context learning to detect and rectify incorrect model predictions. Similarly to [8], [9] investigated the generalization ability of MLLMs in out-of-domain OOD scenarios and studied the effectiveness of in-context learning in bolstering MLLMs' adaptability. In contrast to these studies, which primarily addressed detection and generalization for one type of OOD, our study aimed at detecting OOD samples belonging to several types of OODs simultaneously in the fine-grained manner. We will incorporate this discussion into the revised Related Works section.
>
> [7] Jung et al. Beyond Unimodal: Generalising Neural Processes for Multimodal Uncertainty Estimation. NeuRIPS, 2023.
>
> [8] Huang et al. Machine Vision Therapy: Multimodal Large Language Models Can Enhance Visual Robustness via Denoising In-Context Learning. Preprint, 2023.
>
> [9] Zhang et al., On the Out-Of-Distribution Generalization of Multimodal Large Language Models. Preprint, 2024.

---

### Review · Reviewer_8hZH · 2024-04-12

**Summary Of Contributions:**

This paper presents a novel approach for multi-model anomaly detection, leveraging a methodology akin to CLIP. It employs separate encoders for text and images, and learns a unified representation through joint encoding. Three anomaly scenarios are considered: anomalous images or text, text unrelated to images, and noisy images. Experimental results validate the efficacy of the proposed method, demonstrating its efficiency in anomaly detection.

**Audience:**

Yes

**Claims And Evidence:**

Yes

**Requested Changes:**

The assumption that noisy images represent anomalies might affect the model's out-of-distribution (OOD) generalization. Therefore, conducting experiments specifically designed to evaluate OOD generalization is crucial to gauge the method's robustness.

 Although the method's rationale is well-founded, its novelty appears limited. To enhance novelty, consider investigating the application of zero-shot anomaly detection independently on each modality and subsequently merging the results, akin to methodologies demonstrated in approaches like WinCLIP.

Clarification is needed regarding the selection criteria and rationale for using outliers in training. Additionally, assumptions such as outlier exposure require further elaboration and validation to ensure their reliability and applicability.

 It is imperative to compare the proposed method with single-modality approaches in the experimentation phase. Given the utilization of additional information during inference, it is expected that the proposed method will outperform single-modality approaches by a significant margin.

**Strengths And Weaknesses:**

Strengths:
1. Multi-model anomaly detection is a relatively unexplored field, making it worthwhile to focus on.
2. The paper is well-written and easy to follow.
3. The proposed methods are straightforward yet efficient.

Weaknesses:
1. The assumption of noisy images as anomalies may impact the model's out-of-distribution (OOD) generalization. An OOD generalization experiment is necessary to assess the method's robustness.
2. While the rationale behind the method is sound, its novelty is limited. Exploring the use of zero-shot anomaly detection on each modality and merging the results, as demonstrated in approaches like WinCLIP, could enhance novelty.
3. The method mentions using outliers for training, but it's unclear how these outliers are selected and why. Additionally, assumptions like outlier exposure need further clarification and validation.
4. Experimentation should include comparisons with single-modality approaches. The expectation is that the proposed method should perform significantly better due to its utilization of more information during inference.

---

> ### Author Response · Authors · 2024-05-03
> **Response to Reviewer 8hZH (Part 1/2)**
>
> Q1: The assumption that noisy images represent anomalies might affect the model's out-of-distribution (OOD) generalization. Therefore, conducting experiments specifically designed to evaluate OOD generalization is crucial to gauge the method's robustness.
> --
> **A1**: Per your suggestion, we have evaluated the CLIP model (ViT-B/16) using linear probing for bird species classification on the CUB-200 dataset. Then, we experimentally added Gaussian blur to different percentages of image patches from 10% up to 50%, and reported the Macro-F1 scores to measure the model's classification accuracy for noisy images in Table 1 below. In particular, even when little noise is added (10% of patches), the prediction performance of the CLIP model drops by 6% if half of testing images contain noise, and by 14% if all images are noisy. Additionally, the CLIP model only achieves 60.12% Macro-F1 score on the noisy testing dataset with 50% noisy patches, which is a 22% reduction in performance. Therefore, we concluded that CLIP is not robust to noisy images.
>
> Table 1: OOD generalization evaluation for classification of noisy samples on CUB-200 dataset. The performance is measured using Macro-F1 scores (%).
> |% Noisy Samples \ % Patches|0|10|25|50|
> |-|-|-|-|-|
> |0| **77.62**|_ |_ |_ |
> |25|_ |75.62|74.43|73.58|
> |50|_ |72.78|70.04|69.22|
> |100|_ |66.74|62.40|60.12|
>
> Q2: While the rationale behind the method is sound, its novelty is limited. Exploring the use of zero-shot anomaly detection on each modality and merging the results, as demonstrated in approaches like WinCLIP, could enhance novelty.
> --
> **A2**: We would like to point out the big difference between our work and WinCLIP. Our work aims to detect three different OOD scenarios (_misaligned_, _new-domain_, and _noisy_) in a fine-grained manner, while WinCLIP [1] can only detect one specific OOD scenario where an image from a category (e.g., bird) contains class-specific anomalies such as noise, flaws, damages, etc. Specifically, zero-shot WinCLIP detects anomalies by matching multi-scale CLIP image embeddings (from image windows of varying sizes) with text prompts such as "a blurry photo of a bird". In contrast, our proposed method utilizes contrastive learning for capturing the alignments between input images and their corresponding detailed textual descriptions, and a binary OOD classifier for significantly improved detection performance for three scenarios simultaneously.
>
> Moreover, our experimental results in Tables 2-4 demonstrated that our method outperforms WinCLIP on all three datasets. Additionally, WinCLIP only slightly outperforms other zero-shot baselines including MCM-OOD and GL-MCM-OOD for the _noisy_ scenario.
>
> Table 2: Evaluation on CUB-200 averaged over 3 random seeds.
> |CUB-200| Misaligned+ID||New-domain+ID||Noisy+ID||All Scenarios+ID||
> |-|-|-|-|-|-|-|-|-|
> || AUROC↑|FPR95↓|AUROC↑|FPR95↓|AUROC↑|FPR95↓|AUROC↑|FPR95↓|
> |MCM-OOD |47.16 |94.76| 81.42| 62.72| 51.62 |93.82 |60.07 |90.52|
> |GL-MCM-OOD| 46.90 |96.33| 79.88| 72.50 |49.60 |96.25| 58.80| 94.42|
> | WinClip | 54.91         | 89.55  | 66.13         | 67.26  | 69.62    | 60.48  | 64.87            | 75.96  |
> | Ours    | **91.71**         | **28.47**  | **99.08**         | **3.63**   | **96.73**    | **6.83**  | **95.18**            | **13.98**  |
>
> Table 3: Evaluation on MIMIC-CXR averaged over 3 random seeds.
> |MIMIC-CXR|Misaligned+ID||New-domain+ID||Noisy+ID||All Scenarios+ID||
> |-|-|-|-|-|-|-|-|-|
> ||AUROC↑|FPR95↓|AUROC↑|FPR95↓|AUROC↑|FPR95↓|AUROC↑|FPR95↓|
> |MCM-OOD| 53.43 |96.89 |50.92 |93.54| 47.19 |96.01 |50.49 |95.55|
> |GL-MCM-OOD| 54.15| 95.54| 57.05| 90.05| 47.15| 96.29| 52.78|94.41|
> | WinClip | 51.99|94.32|74.22|81.43| 52.32| 93.05| 62.15|92.21|
> | Ours | **89.02** | **37.66**| **99.81**| **0.47** | **96.95**| **8.03**| **94.48** | **15.39**|
>
> Table 4: Evaluation on COCO averaged over 3 random seeds.
> |COCO|Misaligned+ID||New-domain+ID||Noisy+ID||All Scenarios+ID||
> |-|-|-|-|-|-|-|-|-|
> ||AUROC↑|FPR95↓|AUROC↑|FPR95↓|AUROC↑|FPR95↓|AUROC↑|FPR95↓|
> |MCM-OOD |40.86| 97.99| 71.15| 68.05 |47.39| 96.15| 53.13| 95.42|
> |GL-MCM-OOD| 35.44| 99.73 |86.29| 46.40| 47.66 |96.53| 56.50| 98.40|
> | WinClip | 52.13         | 90.96  | 59.37         | 88.61  | 52.87    | 86.09  | 54.64            | 88.78  |
> | Ours    | **99.78**         |**1.14**   | **99.88**         | **0.88**   | **99.08**    | **2.89**   | **99.58**            | **1.54**   |
>
> [1] Jeong et al. Winclip: Zero-/few-shot anomaly classification and segmentation. CVPR, 2023.

---

> ### Author Response · Authors · 2024-05-03
> **Response to Reviewer 8hZH (Part 2/2)**
>
> Q3: The method mentions using outliers for training, but it's unclear how these outliers are selected and why. Additionally, assumptions like outlier exposure need further clarification and validation.
> --
> **A3**: During training, we randomly selected 1% of the training data and replace them with OOD samples for each scenario (_misaligned_, _new-domain_, and _noisy_) as demonstrated in Table 5 below, resulting in a total of 3% OOD samples and 97% of ID samples. To enable the model to learn better representations for OOD detection, outlier exposure utilizes an auxiliary outlier dataset under the assumption that it is entirely disjoint from testing OOD data [2]. Alternatively, our proposed method draws upon a tiny collection of randomly selected OOD samples to minimize the overlap between the training and testing OOD data, while significantly improve the detection performance.
>
> Table 5: Summary of Generation Methods for three OOD Scenarios.
> | Scenarios  | Generation Method                                                                                                                                                                     |
> |------------|---------------------------------------------------------------------------------------------------------------------------------------------------------------------------------------|
> | Misaligned | Randomly select pairs of ID images and texts, swap their textual descriptions with textual descriptions of images from different categories, and label these misaligned pairs as OOD. |
> | New-domain | Randomly select OOD pairs of images and texts from another new dataset different from the training data.                                                                              |
> | Noisy      | Randomly select pairs of ID images and texts, add a small number of blurry/whitened patches to each, and label these blurry pairs as OOD.                                             |
>
> [2] Hendrycks et al. Deep Anomaly Detection with Outlier Exposure. ICLR, 2019
>
> Q4: Experimentation should include comparisons with single-modality approaches. The expectation is that the proposed method should perform significantly better due to its utilization of more information during inference.
> --
>
> **A4**: In our experiments, we have compared the proposed method against the single-modality methods, such as CLIP-Energy (energy-based finetuning) and CLIP-BCE (outlier exposure), as shown in Table 2-4 in our manuscript. We can see that our method significantly outperforms these two baselines. Moreover, a prior study [3] conducted a comparison between the representative visual OOD detection approach MSP and their proposed MCM-OOD method, which underscores the efficacy of incorporating multimodal information during inference, leading to improved OOD detection performance.
>
> [3] Ming et al. Delving into out-of-distribution detection with vision-language representations. NeuRIPS, 2022.

---

> > ### Comment · Reviewer_8hZH · 2024-05-30
> > **Thank you to response my comments**
> >
> > Currently, I am convinced with the answeres

---

### Review · Reviewer_QqpD · 2024-05-03

**Summary Of Contributions:**

The goal of this paper is to address multi-modal OOD detection. Specifically, it examines three distinct OOD scenarios: misaligned, new-domain, and noisy. The paper introduces a weakly supervised multi-modal OOD detection model named WOOD, which includes a binary classifier and a contrastive learning module to assess the similarity between various modalities. For the contrastive learning module, the authors use a hinge loss, whereas the binary classifier employs a feature sparsity regularizer to effectively integrate relevant features from multiple modalities. Empirical evaluations demonstrate that the WOOD model provides improvements over existing baselines.

**Audience:**

Yes

**Claims And Evidence:**

Yes

**Requested Changes:**

Major comments

- See weaknesses above.

- The authors use three datasets in the experiments. It is not clear if these are the only datasets used by SOTA methods. Can the authors comment on this?


- I think the following paper is relevant to this work. Could the authors discuss it? How does the proposed approach addresses issues raised in this paper?
	- https://openreview.net/forum?id=HyzCuCV1jH


Minor comments

- "We generate a small number of OOD samples K during training"
	- It is not during the training process that these samples are generated.
-  "we adopt Hinge loss instead of L1 in Eq. 1"
	- Please provide the L1 expression, at least in the appendix. Was it used before in other papers?


- "to obtain the OOD predictions", "the prediction results", etc.
	- what do you mean by predictions? Please be more specific, i.e. probability, label, etc.

- Page 6, circled plus and product are not defined.

- In (7), please give the range for lambda (lambda in [0, 1]?).

- Page 11, "performs the best performance"

- Table 6, why did not not consider more extreme values of lambda (lambda = 0 or 1?)

- The plots in Figure 2 are missing the axis labels which make it difficult to follow.
In Equation 2, there is a missing bracket.

**Strengths And Weaknesses:**

Strengths:

- The paper addresses multi-modal OOD beyond image-text alignment. Two additional challenging scenarios are considered where the images are either from a different domain or are noisy.

- The proposed hinge loss for contrastive learning further helps increase the similarity between text and image of ID samples. Additionally, it constrains the image-text similarity for OOD samples, improving their detection.

- Empirical results demonstrate improvements over various baselines on three benchmark datasets.

Weaknesses:

- Experimental Setup:

In Section 5.1, it is mentioned that misaligned samples are generated by replacing the text of an image-text pair with text from another category. However, while discussing the main results in Section 5.4, it is said that the misaligned samples have only fine-grained differences such as colour or appearance. How do you explain this discrepancy?

- Baselines:

The baseline CLIP-BCE has a very similar objective to the proposed mehtod, where they minimize the similarity between image-text pairs for ID samples and maximize it for OOD samples. However, in the results, it seems that this baseline performs the worst at detecting the misaligned OOD samples for which it is trained rather than detecting OOD samples that are noisy or belong to a new domain. How do you explain this? This would allow us to understand the shortcomings of CLIP-BCE and the strengths of WOOD.


- Readability:

The paper is hard to follow in places. For example, the encoders introduced while discussing the binary classifier are not well defined. It is only in the discussion of the model setup, that the readers are informed that the encoders are single layer MLPs.

Furthermore, without knowing that the input and output of the encoders are the same, it is difficult to understand how the weights can be multiplied by the features. Mentioning the mapping or the dimensions could further improve the readability.

- Additional Information:

The scoring metric in Equation 8 is not clear. To the best of my understanding, the value of $P_{cl}$ does not have a range of 0 to 1 as it is obtained by applying sigmoid on the cosine similarity, whose value lies in the range -1 to 1. Thus, the scoring metric would also never be 1. Please give more details and provide the range in which the values of the scoring metric lie.


- Extended Ablations:

While discussing the results from the ablation studies, the authors refer to the three OOD scenarios and discuss how each is addressed by a specific choice. The authors should provide performance for each OOD scenario separately rather than just the mean performance.


- Experimental Results:

In the paper it is mentioned that the results are averaged over three random seeds but the authors do not provide standard errors.

---

> ### Author Response · Authors · 2024-05-20
> **Response to Reviewer QqpD (Part 1/4)**
>
> Q1: In Section 5.1, it is mentioned that misaligned samples are generated by replacing the text of an image-text pair with text from another category. However, while discussing the main results in Section 5.4, it is said that the misaligned samples have only fine-grained differences such as colour or appearance. How do you explain this discrepancy?
> --
> **A1**: We would like to note that the CUB-200 dataset contains 200 categories of bird species. Consequently, when misaligned samples are generated as detailed in Section 5.1, each of the replaced text from another category still describes a bird, which may has very similar features as the bird from the original category. For instance, "this bird has a pointed beak, yellow and black feathers" is replaced by "this bird is white with black and has a very short beak". Therefore, while discussing the OOD detection results in Section 5.4, we highlighted that textual descriptions from the replaced and the original categories may only differ in a fine-grained manner such as mentions of different colors or appearances.
>
> Q2: The baseline CLIP-BCE has a very similar objective to the proposed method, where they minimize the similarity between image-text pairs for ID samples and maximize it for OOD samples. However, in the results, it seems that this baseline performs the worst at detecting the misaligned OOD samples for which it is trained rather than detecting OOD samples that are noisy or belong to a new domain. How do you explain this? This would allow us to understand the shortcomings of CLIP-BCE and the strengths of WOOD.
> --
> **A2**: We would like to point out the big difference between our work and CLIP-BCE. CLIP-BCE formulates the OOD detection problem as a two-class classification for matching an image with two different texts: "a photo of something else", and "a photo of a {c}" where c is the given category of the image. Consequently, its objective aims to maximize the similarity between OOD images and "a photo of something else", and minimize that for ID samples. Intuitively, since CLIP-BCE detects OODs using the cosine similarity score between an image and "a photo of something else", it is suitable for detecting the _new-domain_ and _noisy_ scenarios, but not the _misaligned_ scenario as demonstrated in our experiments. In contrast, our proposed method utilized the contrastive objective to learn the (mis)alignment between each image and its _detailed textual descriptions_, which naturally lends itself to detect _misaligned_ OODs.
>
> Q3: The paper is hard to follow in places. For example, the encoders introduced while discussing the binary classifier are not well defined. It is only in the discussion of the model setup, that the readers are informed that the encoders are single layer MLPs. Furthermore, without knowing that the input and output of the encoders are the same, it is difficult to understand how the weights can be multiplied by the features. Mentioning the mapping or the dimensions could further improve the readability.
> --
> **A3**: We would like to clarify that the encoders introduced while discussing the binary classifier are single layer MLPs utilized for obtaining the sparse weights in the feature sparsity regularizer. First, we used the CLIP image and text encoders to encode input images and texts respectively, then these encoded features are fed into single layer MLPs to yield a sparse weight vectors of the same dimension. Finally, the sparse features are obtained by the element-wise multiplication of the CLIP encoded features and their corresponding weights. Per your suggestion, we have included these details when we introduced the binary classifier in Section 4 in the revised version of the manuscript.

---

> ### Author Response · Authors · 2024-05-20
> **Response to Reviewer QqpD (Part 2/4)**
>
> Q4: The scoring metric in Equation 8 is not clear. To the best of my understanding, the value of does not have a range of 0 to 1 as it is obtained by applying sigmoid on the cosine similarity, whose value lies in the range -1 to 1. Thus, the scoring metric would also never be 1. Please give more details and provide the range in which the values of the scoring metric lie.
> --
> **A4**: We have clarified the scoring metric in Eq. 8 as follows. To obtain the ID probabilities $P\_{cl}$ from the contrastive learning module, we followed previous work [1] which utilizes the tempered sigmoid function $\sigma(S(x\_n, t\_n)/\tau)$ to convert the cosine similarities between image-text pairs into prediction probabilities within the range (0, 1). Note that when $\tau$ is small (e.g., 0.1), the range of $\sigma(S(x\_n, t\_n)/\tau)$ is very close to $(0, 1)$. We have revised Eq. 5 in the manuscript as $P\_{cl}=\sigma(S(x\_n, t\_n)/\tau)$ to reflect these details.
>
> In addition, the ID probabilities from the binary classifier $P\_{bc}$ are also in (0, 1). Hence, when $P\_{cl}$ are multiplied with $P\_{bc}$ and the OOD scores are calculated according to Eq. 8: $P\_{ood} = 1 - P\_{cl}\times P\_{bc}$, the scoring metric $P\_{ood}$ will also be between 0 and 1.
>
> [1] MuMIC: Multimodal Embedding for Multi-label Image Classification with Tempered Sigmoid. AAAI, 2023
>
> Q5: While discussing the results from the ablation studies, the authors refer to the three OOD scenarios and discuss how each is addressed by a specific choice. The authors should provide performance for each OOD scenario separately rather than just the mean performance.
> --
> **A5**: Per your suggestion, we have added the detailed OOD detection results regarding the three OOD scenarios in our ablation study on the main model components in Table 1 below. From this table, we can draw two conclusions: (i) notably, our method incorporating both the binary classifier (BC) and the contrastive learning (CL) components demonstrated superior performance; and (ii) each component exhibits a notable influence on performance, affirming the necessity of their inclusion in our method. We have included these results in Table 5 of the revised manuscript.
>
> Table 1: Impact of ablating the binary classifier (BC) and the contrastive learning (CL) components from the proposed WOOD method.
> |    Methods    | Misaligned+ID |           | New-domain+ID |          |  Noisy+ID |          | All Scenarios+ID |           |
> |-|-|-|-|-|-|-|-|-|
> |               |     AUROC↑    |   FPR95↓  |     AUROC↑    |  FPR95↓  |   AUROC↑  |  FPR95↓  |      AUROC↑      |   FPR95↓  |
> |  **CUB-200**  |               |           |               |          |           |          |                  |           |
> | WOOD w/o BC   |         90.84 |     31.67 |         86.94 |    40.08 |     51.19 |    94.67 |            76.32 |     55.47 |
> | WOOD w/o CL   |         49.26 |     93.25 |         97.62 |     7.25 |     88.74 |    33.83 |            78.52 |     13.98 |
> | WOOD          |     **91.71** | **28.47** |     **99.08** | **3.63** | **96.73** | **6.83** |        **95.18** | **13.98** |
> | **MIMIC-CXR** |               |           |               |          |           |          |                  |           |
> | WOOD w/o BC   |         86.75 |     51.52 |         86.75 |    48.50 |     51.52 |    94.48 |            72.58 |     68.43 |
> | WOOD w/o CL   |         68.34 |     85.91 |         99.32 |     3.39 |     51.21 |    95.00 |            73.15 |     61.42 |
> | WOOD          |     **89.02** | **37.66** |     **99.81** | **0.47** | **96.95** | **8.03** |        **94.48** | **15.39** |
> |    **COCO**   |               |           |               |          |           |          |                  |           |
> | WOOD w/o BC   |         99.73 |      1.60 |         90.43 |    27.12 |     53.59 |    94.02 |            81.25 |     41.24 |
> | WOOD w/o CL   |         77.90 |     80.91 |         99.48 |     2.02 |     96.80 |    10.22 |            91.39 |     31.05 |
> | WOOD          |     **99.78** |  **1.14** |     **99.88** | **0.88** | **99.08** | **2.89** |        **99.58** |  **1.54** |

---

> ### Author Response · Authors · 2024-05-20
> **Response to Reviewer QqpD (Part 3/4)**
>
> Q6: In the paper it is mentioned that the results are averaged over three random seeds but the authors do not provide standard errors.
> --
> **A6**: According to your suggestion, we have provided the mean and standard deviations (separated by $\pm$) over 3 random seeds for the AUROC and FPR95 scores for the OOD detection performance on the CUB-200 dataset in Table 2 below. We have also included the standard deviations in Tables 2-4, which illustrate the detection performance of our proposed method compared to the baselines on 3 datasets, in the revised manuscript.
>
> Table 2: Performance comparison of different methods for OOD detection on CUB-200 dataset averaged over three random seeds. The standard deviations across random seeds are reported following ±. Higher AUROC and lower FPR95 indicate better performance.
>
> |      Methods      | Misaligned+ID |              | New-domain+ID |              |   Noisy+ID   |              | All Scenarios+ID |              |
> |-|-|-|-|-|-|-|-|-|
> |                   |     AUROC↑    |    FPR95↓    |     AUROC↑    |    FPR95↓    |    AUROC↑    |    FPR95↓    |      AUROC↑      |    FPR95↓    |
> |     **Zero-shot**     |               |              |               |              |              |              |                  |              |
> | MCM-OOD           |  47.16 ± 2.74 | 94.76 ± 1.26 |  81.42 ± 0.17 | 62.72 ± 1.01 | 51.62 ± 2.68 | 93.82 ± 5.79 |     60.07 ± 0.41 | 90.52 ± 1.29 |
> | GL-MCM-OOD        |  46.90 ± 1.37 | 96.33 ± 0.38 |  79.88 ± 1.07 | 72.50 ± 1.09 | 49.60 ± 3.91 | 96.25 ± 0.90 |     58.80 ± 2.11 | 94.42 ± 1.53 |
> |     **Finetuned**     |               |              |               |              |              |              |                  |              |
> | CoOp-MCM          |  48.61 ± 1.49 | 95.76 ± 0.67 |  90.30 ± 0.90 | 48.83 ± 2.02 | 52.83 ± 3.30 | 95.55 ± 0.68 |     63.91 ± 1.88 | 94.11 ± 0.77 |
> | NPOS-MCM          |  49.46 ± 1.76 | 94.58 ± 1.51 |  52.34 ± 2.69 | 92.67 ± 1.81 | 46.71 ± 0.64 | 93.83 ± 1.01 |     49.50 ± 0.96 | 94.08 ± 1.38 |
> | CLIP-N            |  49.96 ± 2.36 | 96.00 ± 1.80 |  80.21 ± 4.86 | 83.33 ± 9.75 | 49.64 ± 0.64 | 95.17 ± 1.51 |     59.94 ± 1.74 | 95.17 ± 2.75 |
> | LoCoOp            |  49.25 ± 2.68 | 95.92 ± 1.13 |  91.72 ± 1.66 | 42.58 ± 3.59 | 54.20 ± 3.66 | 92.92 ± 0.14 |     65.05 ± 2.62 | 91.90 ± 0.98 |
> | **Weakly-supervised** |               |              |               |              |              |              |                  |              |
> | CLIP-Energy       |  49.45 ± 1.43 | 95.25 ± 2.05 |  78.51 ± 1.54 | 83.50 ± 4.11 | 50.08 ± 4.36 | 95.67 ± 0.63 |     59.35 ± 2.81 | 93.58 ± 0.72 |
> | CLIP-BCE          |  50.37 ± 3.25 | 94.83 ± 0.29 |  88.61 ± 3.16 | 29.50 ± 5.25 | 95.10 ± 0.39 | 15.67 ± 3.13 |     78.08 ± 1.87 | 46.87 ± 2.49 |
> | Ours              |  **91.71 ± 0.53** | **28.47 ± 1.25** |  **99.08 ± 0.92** |  **3.63 ± 1.18** | **96.73 ± 0.86** |  **6.83 ± 1.26** |     **95.18 ± 0.25** | **13.98 ± 0.60** |
>
> Q7: The authors use three datasets in the experiments. It is not clear if these are the only datasets used by SOTA methods. Can the authors comment on this?
> --
> **A7**: We would like to point out that existing works in single-modality OOD detection have only conducted experiments on image-only datasets such as ImageNet, Places, Texture, etc., or text-only datasets such as Newsgroup, SST-2, etc. Moreover, prior works which utilized the image and text modalities for detecting visual OODs only constructed the corresponding text for each image by specifying its given category in text prompts like "a photo of a {category}" (i.e., these works only used images and and their category labels). Since our study aimed at multimodal OOD detection on datasets containing pairs of images and the entire captions of images, which are much longer and more detailed than the category labels, we selected three real-world datasets that satisfy this requirement: CUB-200, MIMIC-CXR, and COCO.

---

> ### Author Response · Authors · 2024-05-20
> **Response to Reviewer QqpD (Part 4/4)**
>
> Q8: I think the following paper is relevant to this work. Could the authors discuss it? How does the proposed approach addresses issues raised in this paper?
> --
> **A8**: We have discussed the suggested related work as follows. [2] considered data augmentation as a hyperparameter for generating pseudo-anomalies, and investigated its effectiveness for image-based self-supervised anomaly detection (SSAD). Consequently, they highlighted that selecting data augmentation hyperparameters that align with true anomalies is critical to the success or failure of existing SSAD techniques. However, this work primarily studied the detection of semantic class anomalies where images come from anomalous or unknown classes, and provided no solution to address the hyperparameter selection problem for data augmentation. In contrast, our work focused on multimodal OOD detection for several scenarios of OODs such as misalignment, new domain, and noise simultaneously. To this end, we utilized weak supervision on a small collection of randomly generated multimodal OOD samples representative of these scenarios to improve the detection accuracy. We have discussed this work in the Related Work section in our revised manuscript.
>
> [2] Yoo et al. Data Augmentation is a Hyperparameter: Cherry-picked Self-Supervision for Unsupervised Anomaly Detection is Creating the Illusion of Success. TMLR, 2023.
>
> Q9: "We generate a small number of OOD samples K during training". It is not during the training process that these samples are generated.
> --
> **A9**: We would like to clarify that we generated a small number of OOD samples for each training dataset using a random seed at the beginning of each training run.
>
> Q10: "We adopt Hinge loss instead of L1 in Eq. 1". Please provide the L1 expression, at least in the appendix. Was it used before in other papers?
> --
> **A10**: We have provided the $\mathcal{L}\_1$ expression as Eq. 1 below (also Eq. 1 in the submission). In particular, the first term is similar to the CLIP contrastive loss [3], which minimizes the cross entropy of the cosine similarities between $N-K$ ID image-text pairs. Regarding the second term, we reversed the sign of the CLIP contrastive loss for $K$ OOD image-text pairs to maximize the cross entropy of their cosine similarities.
>
> Equation 1: Modified CLIP contrastive loss.
> $$
> \mathcal{L}\_{1} = -\frac{1}{N-K}\sum\_{n=1}^{N-K} \log\frac{e^{S\_{id}(x\_n^+,t\_n^+)}}{\sum\_{i=1}^{N-K} e^{S\_{id}(x\_n^+,t\_i^+)}} + \frac{1}{K} \sum\_{k=1}^{K} \log \frac{e^{S\_{ood}(x\_k^-,t\_k^-)}}{\sum\_{i=1}^{K} e^{S\_{ood}(x\_k^-,t\_i^-)}}
> $$
>
> [3] Radford et al. Learning transferable visual models from natural language supervision. PMLR, 2021.
>
> Q11: "to obtain the OOD predictions", "the prediction results", etc. What do you mean by predictions? Please be more specific, i.e. probability, label, etc.
> --
> **A11**: We have included further clarification with additional details as follows. Both "OOD predictions" and "prediction results" in Page 6 should be "ID probabilities", resulted from from the contrastive learning module and/or the binary classifier.
>
> Q12: Page 6, circled plus and product are not defined.
> --
> **A12**: We have defined the circled plus ($\oplus$) and product ($\otimes$) notations as follows. $\oplus$ indicates the concatenation of two vector representations, and $\otimes$ denotes the element-wise multiplication of two vector representations. We have also included these details in the revised manuscript.
>
> Q13: In Eq. 7, please give the range for lambda (lambda in [0, 1]?).
> --
> **A13**: We would like to clarify that the range for $\lambda$ in Eq. 7 is in $[0, 1]$. We have added this detail to the revised manuscript.
>
> Q14: Page 11, "performs the best performance"
> --
> **A14**: Thank you for your careful proofreading. We have fixed the typos as follows. The “performs the best performance” should be “perform the best”.
>
> Q15: Table 6, why did not not consider more extreme values of lambda (lambda = 0 or 1?)
> --
> **A15**: We would like to note that we have considered these extreme values of $\lambda$ when examining the effect of main model components in the first ablation study, as illustrated in Table 5 in the manuscript. In particular, when $\lambda= 0$, the proposed method only makes use of the contrastive learning module. Additionally, when $\lambda = 1$, only the binary classifier is utilized. We have included these clarifications in the details of our ablation studies.
>
> Q16: The plots in Figure 2 are missing the axis labels which make it difficult to follow. In Equation 2, there is a missing bracket.
> --
> **A16**: Per your suggestion, we have added the axis labels in Figure 2, where the x-axis denotes the ID probabilities, and the y-axis indicates the data sample count for 100 bins of the ID probabilities. Additionally, we have corrected the missing bracket in Eq. 2.

---

> > ### Comment · Reviewer_QqpD · 2024-05-29
> >
> > I thank the authors for their thorough response. They have addressed my comments satisfactorily.

---

> > ### Comment · Reviewer_QqpD · 2024-05-29
> >
> > I thank the authors for their thorough response. They have addressed my comments satisfactorily.

---

### Decision · Action_Editor_fb5x · 2024-06-14

**Recommendation:** Accept as is

**Comment:**

This paper proposes a method for multi-model out-of-distribution detection. Unlike previous work, various forms of "out-of-distributionness" are considered, and the empirical results are good. Reviewers support acceptance, and I do as well.

**Audience:**

Reviewers unanimously agree this paper will be of interest to a subset of TMLR's audience.

**Claims And Evidence:**

Reviewers unanimously agree that the authors provide evidence to support the claims they make.